# Reconstructing the transport cycle in the sugar porter superfamily using coevolution-powered machine learning

**Darko Mitrovic[1], Sarah E McComas[1,2], Claudia Alleva[1,2], Marta Bonaccorsi[1,2], David Drew[2]\*, Lucie Delemotte[1]\***

[1]Department of Applied Physics, Science for Life Laboratory, KTH Royal Institute of Technology, Stockholm, Sweden; [2]Department of Biochemistry and Biophysics, Science for Life Laboratory, Stockholm University, Stockholm, Sweden

**Abstract** Sugar porters (SPs) represent the largest group of secondary-active transporters. Some members, such as the glucose transporters (GLUTs), are well known for their role in maintaining blood glucose homeostasis in mammals, with their expression upregulated in many types of cancers. Because only a few sugar porter structures have been determined, mechanistic models have been constructed by piecing together structural states of distantly related proteins. Current GLUT transport models are predominantly descriptive and oversimplified. Here, we have combined coevolution analysis and comparative modeling, to predict structures of the entire sugar porter superfamily in each state of the transport cycle. We have analyzed the state-specific contacts inferred from coevolving residue pairs and shown how this information can be used to rapidly generate free-energy landscapes consistent with experimental estimates, as illustrated here for the mammalian fructose transporter GLUT5. By comparing many different sugar porter models and scrutinizing their sequence, we have been able to define the molecular determinants of the transport cycle, which are conserved throughout the sugar porter superfamily. We have also been able to highlight differences leading to the emergence of proton-coupling, validating, and extending the previously proposed latch mechanism. Our computational approach is transferable to any transporter, and to other protein families in general.

\*For correspondence:
david.drew@dbb.su.se (DD);
lucie.delemotte@scilifelab.se
(LD)

## Editor's evaluation

This important work proposes a novel approach, based on co-evolution analysis, machine-learning protocols, and molecular dynamics simulations, to predict structures and energetics of the main states of the alternating access cycle of a family of membrane transporters, the sugar porters. The approach is compelling, especially the application of co-evolution and Alphafold to generate accurate models in different conformational states of a given protein, and will be of interest to the membrane transport and computational modeling communities.

## Introduction

Due to the importance of glucose and other monosaccharides for cell metabolism (*Koepsell, 2020*), sugar porters (SPs) are the largest and widest spread family of small molecule transporters across all kingdoms of life (*Drew et al., 2021*; *Reddy et al., 2012*). In mammals, the sugar porters are referred to as glucose transporters (GLUTs) belonging to the Solute Carrier Family 2 A (*Wang et al., 2020*; *César-Razquin et al., 2015*; *Holman, 2020*). Human has 14 different GLUT isoforms, and each isoform has a distinct pattern of tissue distribution, gene regulation, substrate preference, and kinetic properties

(*Holman, 2020*; *Mueckler and Thorens, 2013*; *Huang and Czech, 2007*). GLUT1, for example, is distributed in a wide range of tissues, including the blood–brain barrier, and is essential for glucose transport into the brain (*Koepsell, 2020*; *Holman, 2020*), whereas GLUT4 is mostly localized in skeletal muscle and adipose tissue, and is the major insulin-stimulated glucose transporter (*Huang and Czech, 2007*). GLUT5 is the only member specific to fructose and is the major route for its intestinal absorption (*Rand et al., 1993*; *Douard and Ferraris, 2008*). In plants, fungi, bacteria, and parasites the sugar porter family has likewise expanded into a large number of different isoforms, providing essential and niche roles in the uptake of D-glucose and other monosaccharides. Plants, for example, express specific sugar porter isoforms for seed, fruit, and pollen production (*Niño-González et al., 2019*) and yeast have 20 different hexose transporters with various kinetics (*Boles and Hollenberg, 1997*), which are targeted for biofuel production (*Wang et al., 2016*). Parasites, such as *Plasmodium falciparum* rely on sugar transport to infect their host and reproduce (*Woodrow et al., 1999*; *Blume et al., 2011*), resorting to a promiscuous sugar porter, which can take up a variety of sugars (*Qureshi et al., 2020*). Understanding the molecular basis for sugar transport is thus a fundamental question in biology, with important medical and biotechnological applications.

Sugar porters belong to the Major Facilitator Superfamily (MFS) (*Drew et al., 2021*; *Reddy et al., 2012*) and are defined by an N- and a C-terminal bundle of six TMs, which are connected by a cytosolic loop that is made up of three to four intracellular helices (ICH) (*Drew et al., 2021*; *Drew and Boudker, 2016*). The sugar porters can be subclassified by a sequence motif (*Maiden et al., 1987*), which structures have shown corresponds to an intracellular salt-bridge network that selectively stabilizes the outward-facing state (*Drew and Boudker, 2016*; *Quistgaard et al., 2013*; *Deng et al., 2015*; *Nomura et al., 2015*). Constructing a transport cycle requires assembling five different conformational states along the transport cycle: outward open, outward occluded, occluded, inward occluded, and inward open (*Figure 1A–C*; *Drew et al., 2021*; *Drew and Boudker, 2016*). Globally, sugar porters operate according to the rocker-switch alternating-access mechanism (*Drew et al., 2021*; *Drew and Boudker, 2016*). To transport the substrate, the substrate-adjacent TM7b helix first needs to bend, before breaking into two halves (*Figure 1B*, left), after which the bundles rock and TM10 rearranges to become more perpendicular to the membrane plane (*Figure 1B*, middle). Lastly, the intracellular gate opens via the rearrangement of occluding contacts between TM4 and TM10 (*Figure 1B*, right). Although the two bundles are structurally similar, D-glucose is not coordinated evenly, but almost entirely by residues located in the C-terminal bundle (*Qureshi et al., 2020*; *Deng et al., 2014*; *Deng et al., 2015*). As a consequence, half helices TM7b and TM10b in the C-terminal bundle are thought to undergo local rearrangements to control access to the sugar-binding site from the outside and inside, respectively (*Drew et al., 2021*; *Drew and Boudker, 2016*; *Deng et al., 2015*; *Nomura et al., 2015*; *Figure 1B*). Whilst the sugar porter family is structurally the best characterized out of all MFS transporters (*Drew et al., 2021*), there is still not a single sugar porter that has structures determined in all of these different states. Indeed, some conformations, such as the occluded and inward-occluded states (*Qureshi et al., 2020*; *Wisedchaisri et al., 2014*; *Jiang et al., 2020*), have only been experimentally resolved in one isoform. Due to the paucity of structural information, transport models are currently descriptive, lacking rigorous validation. Consequently, it is still unclear if the single snapshots represent physiologically relevant conformations along the transport cycle, and if unique structural features are transferable to more distantly related members.

Despite low sequence identity, structures of distantly related sugar porters show a high degree of structural conservation as compared to many other types of MFS transporters (*Drew et al., 2021*; *Qureshi et al., 2020*). Using the 20 available experimental structures spread across all major conformations (*Drew et al., 2021*, *Figure 1C*) we hypothesized that we should be able to pinpoint co-evolving residues pairs specific to each state, and then use these contacts for structure prediction. Notably, we reasoned that this approach would be superior to either (1) homology modeling, which is intrinsically biased toward the captured structure, regardless of whether the features of the template are actually transferable across isoforms or not, or (2) structures generated by coevolution only, which may settle on a conformational state that satisfies the contacts from each of the multiple states yet fails to capture a physiological conformation.

Here, we present a novel approach for state-specific structure prediction. In essence, we have used experimental structures determined in each of the different conformations to train a neural network to identify state-specific contacts. After filtering these contacts using coevolution information, we

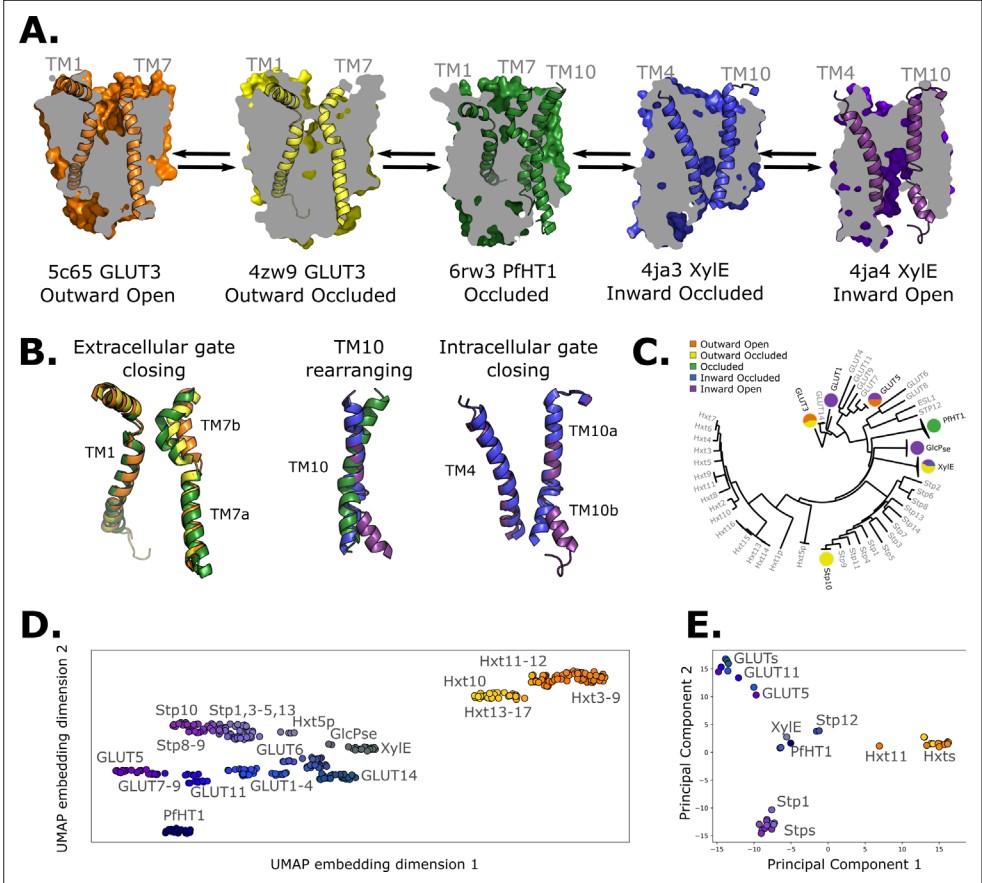

**Figure 1.** Available structural and evolutionary information. (**A**) Clipped surfaces of representative structures of each functional state arranged according to the conformational cycle of sugar transporters (SPs). Note the alternating access to the intra- and extracellular solvents throughout the cycle. The gating helices TM7 and TM10 are shown in their cartoon representations, as well as their partners TM1 and TM4. (**B**) Cartoon representations of the primarily changing molecular features over the conformational cycle. First, the TM7b bends, then breaks when transition from the outward open to the occluded state through the outward-occluded state. In the rocker-switch motion, the TM10 helix rearranges to accommodate the rocking motion, after which the intracellular gate opens to form the inward open state. (**C**) The phylogenetic tree of the SP family, based on a multiple sequence alignment calculated by the MUSCLE algorithm. Tree generated using iTOL. The available experimental structures for each branch are highlighted as colored circles in the proportions that they appear in. The subfamilies represented in this tree are the mammalian sugar transporters (GLUTs), the bacterial and parasitic outliers (*Pf*HT1, XylE, and GlcP$_{se}$), the plant SPs, and the fungal hexose transporters (HXTs). Sequences were retrieved from UniProtKB. (**D**) A 2D embedding of the SP family in sequence space, using as similarity measure scores from the BLOSUM62-derived distance matrix. The embedding displays the phylogenetic distance between different branches of the sugar transporter family and are labeled as dots, colored according to their phylogenetic proximity. (**E**) Principal component analysis (PCA) projection of the available AlphaFold2 models in the SP family using as features residue–residue distances. SPs cluster according to phylogenetic proximity rather than conformational state.

have then used them to apply biases to AlphaFold2 models (*Jumper et al., 2021*) and driven them toward the various conformational states along the functional cycle. Next, we have combined these contacts into collective variables (CVs) to use in enhanced sampling molecular dynamics (MD) simulations (*Harpole and Delemotte, 2018*; *Yang et al., 2019*), and computed the free-energy landscape of the fructose transporter GLUT5 (*Nomura et al., 2015*; *Figure 2A*). Finally, with a set of conserved state-specific contacts defined, we were able to pinpoint both contacts that govern the conformational cycle across the sugar porter family, and a series of transporter-specific interactions that control conformational cycling of each of the sugar porter subfamilies. In particular, we concentrate on deciphering the molecular and evolutionary determinants of sugar–proton-coupled symport.

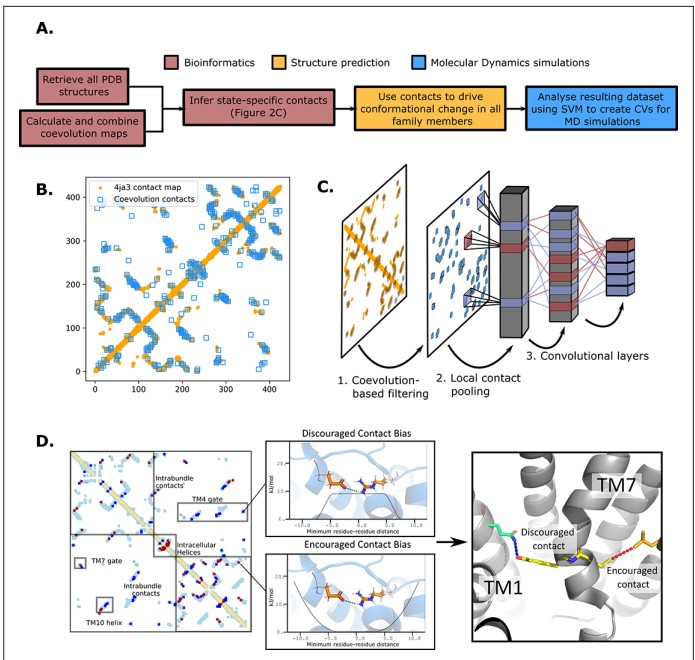

**Figure 2.** The state-specific structure prediction method. (**A**) Summarized pipeline for obtaining a description of the conformational change in sugar transporters, colored by which technique is predominantly used. (**B**) An overlay of the top 200 coevolving residue pairs (blue) and the experimental contact map of the bacterial XylE structure 4ja3 in an inward-occluded state (orange). The secondary structure contacts from the coevolutionary analysis were omitted. (**C**) The architecture of the neural network used to train to predict conformational states from contacts between residue pairs. The first step consists in filtering out the low-coevolving contacts by the coevolution-based filtering, after which a restricted convolutional layer works to pool together close contacts in 4 × 4 grids, to account for the fact that contacts between adjacent positions often serve similar functions. Then, two unrestrained convolutions connect the aforementioned layer to the hidden layer and to the five-dimensional output layer, each node representing a single conformational state. (**D**) The bias application. The highly encouraged and discouraged contacts are translated into Ambiguous Constraint and Multi Constraint type biases in the RosettaMP fastrelax module, with the functional forms displayed in the middle panel. Using as input AlphaFold2 models of each member in the family, the biases are applied and the energy is minimized in a Monte-Carlo fashion, repeated to generate 100 models. The final structure is chosen as the model with the lowest energy score value.

## Results and discussion

### Inferring coevolving conserved residue pairs across all sugar porters

AlphaFold2 structural models of the various sugar porters are thought to collectively capture all major conformational states (*Del Alamo et al., 2022*). However, deterministic prediction of a given state is not possible. Therefore, as expected, principal component analysis (PCA) based on pair-wise interactions reveals that sugar porters cluster within their individual subfamilies rather than according to the conformational state that happens to be predicted by the structure prediction method (*Figure 1D, E*). To be able to rationally build models of individual sugar porters in each of the states in the transport cycle, we thus need to steer AlphaFold2 generated models, or experimentally captured structures, when applicable, into alternative states. We achieve this by using state-specific contacts derived from coevolution analysis (*Figure 2A*).

To do so, we first generated a representative sequence alignment by aligning ~1000 sequences from each of the evolutionarily distant sugar porter relatives belonging to mammalian sugar transporters (GLUTs), the bacterial and parasitic transporters (*Pf*HT1 and XylE), the plant sugar transporters (SPs), and the fungal hexose transporters (HXT), separately (*Figure 1C*). These sugar porter family members were selected because they are either part of a major evolutionary branch of the sugar porter family or are functionally distinct and structurally characterized (as is the case for *Pf*HT1 or XylE, see Methods for details). We then used the resulting multiple sequence alignments (MSAs) as input for direct coupling analysis (DCA) to generate coevolution maps (*Morcos et al., 2011*). The coevolution

**Table 1.** Fraction of encouraged and discouraged contacts for each predicted state shown in *Figure 3*.

The fractions were calculated within the helical bundles (intrabundles 1 and 2), defined as spanning TM1–TM6 and TM7–TM12, respectively. Additionally, interbundle contacts between these two bundles were defined, along with the intracellular helix contacts (ICH1–5). The plus and minus signs denote the encouraged and discouraged contacts, respectively. The percentages concern the fraction of these contacts over all observed contacts in all members of the training set (all experimentally resolved structures).

| | Intrabundle 1 | | Intrabundle 2 | | Interbundle | | ICH | |
|---|---|---|---|---|---|---|---|---|
| | + | − | + | − | + | − | + | − |
| Out Open | 3.6% | 0.58% | 3.34% | 0.58% | 0.19% | 0.17% | 0.09% | 0.12% |
| Out Occ | 3.6% | 0.24% | 3.67% | 0.25% | 0.12% | 0.18% | 0.09% | 0.12% |
| Occ | 3.88% | 0.03% | 3.57% | 0.46% | 1.0% | 0.18% | 0.09% | 0.12% |
| In Occ | 3.64% | 0.31% | 3.40% | 0.77% | 0.2% | 0.16% | 0.09% | 2.9% |
| In Open | 3.52% | 0.27% | 3.64% | 0.33% | 0.09% | 0.20% | 1.12% | 3.0% |

maps generated from separately aligned MSAs were subsequently combined into a global coevolution map, filtering out contacts that were predominantly transporter-specific (see Methods). As illustrated for the proton-coupled xylose transporter XylE (*Figure 2B*), there is an extensive overlap between the MSA-derived coevolution contact maps and experimental inward-facing structural contacts. Nevertheless, several coevolving pairs do not correspond to interactions found in the structure. Overall, we estimate that ~10% of the top 500 coevolving pairs represent contacts not formed in available crystal structures, presumably representing contacts forming in other conformational states (see Methods).

## Extracting state-dependent coevolving residue pairs from experimental structures

To determine which coevolving pairs correspond to contacts that are formed or broken in the 20 structures of different functional states, we trained a convolutional neural network (CNN) to classify the main five conformational states using as input contact maps from experimentally determined structures filtered by coevolution scores (*Figure 2C*). The CNN architecture was designed to avoid redundancy of contacts between neighboring residues while allowing residue pairs to be in contact in several functional states (Methods). Layer-wise relevance backpropagation (LRP) was then performed (*Bach et al., 2015*) on all five output classes separately, identifying contacts that are characteristically present or absent for each conformation. We termed these contacts encouraged and discouraged contacts, respectively (*Figure 2D*).

Interestingly, encouraged and discouraged contacts were not confined to interbundle contacts and are instead spread throughout the entire structure (*Table 1*). They also displayed strong state dependency (*Figure 3*). Because of the low number of experimental structures available, training of the neural network carried a substantial risk of overfitting. We thus sought to validate the results by making sure that the state-specific contacts identified were indeed in the expected regions. Starting from the outward-facing conformation, we observe the expected encouraged contacts of interbundle salt bridges formed between TM4–TM10b and TM5–TM8 helices. In contrast, we observe a strong signal for the discouraged contacts between TM1 and the extracellular gate TM7b, which are known to come together during rocker-switch transition into the inward-facing states (*Drew et al., 2021*; *Nomura et al., 2015*). In the occluded structure of *Pf*HT1 (*Qureshi et al., 2020*), the extracellular gate TM7b had moved fully inwards and transition into a broken helix at the point closest to TM1. Furthermore, mutations in TM1 were found to be just as critical for transport as those in TM7b (*Qureshi et al., 2020*), indicating that TM1 and TM7b interactions might be important in driving formation of the occluded state. Indeed, we observe a robust signal for encouraged contacts between TM1 and TM7b, which is present as early as the occluded state (*Table 2*), confirming that TM7b and the interactions with TM1 are coevolving to enable attainment of the occluded conformation. As expected for an intermediate state, the occluded state has the maximal number of encouraged contacts. One

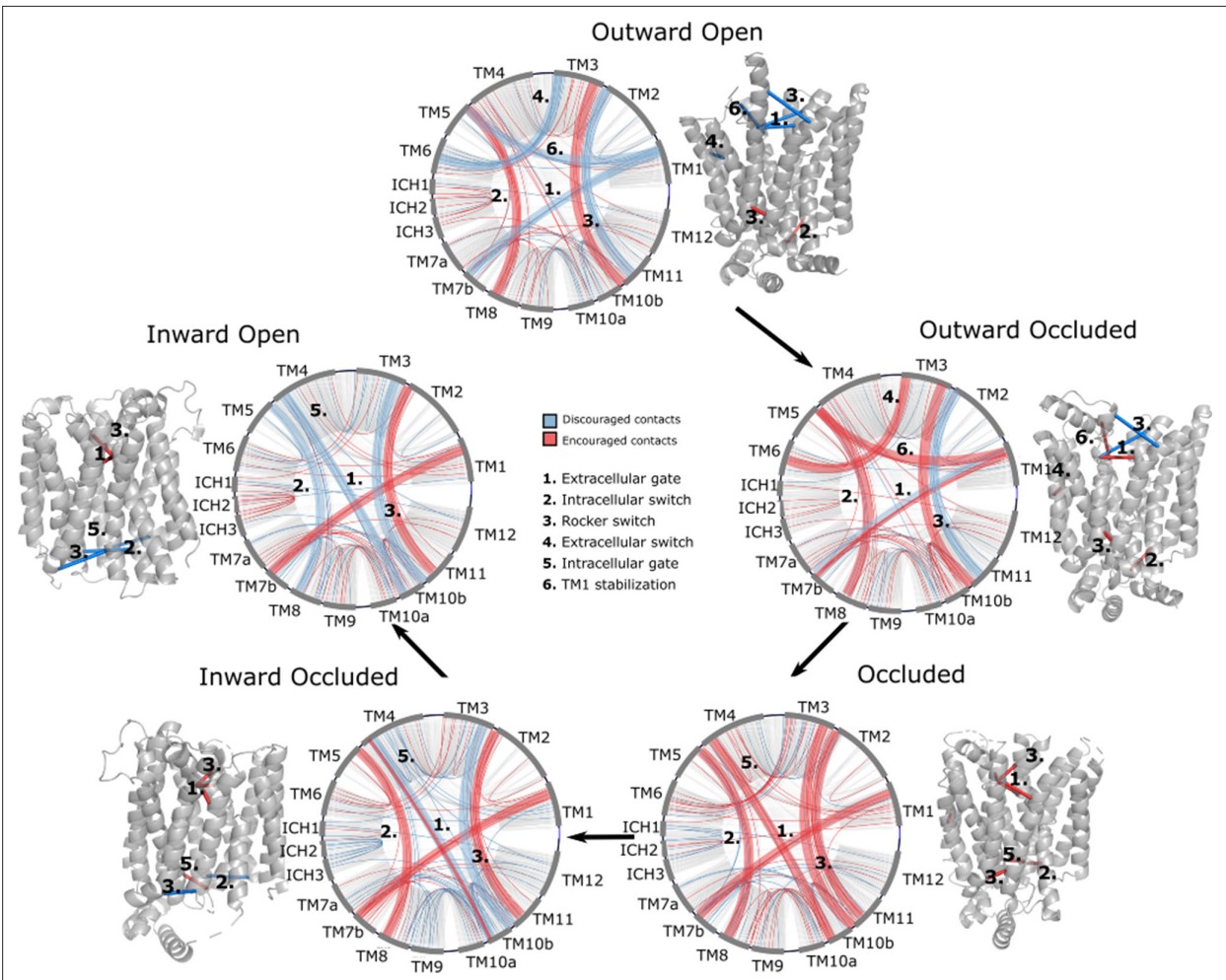

**Figure 3.** Network representations of the state-specific contact maps from the layer-wise relevance backpropagation (LRP) analysis of the trained neural network. Nodes are labeled by the helix they are a part of, and the edges are colored by their sign in the LRP – blue represents discouraged contacts, and red represents encouraged contacts. Consensus contact maps of all states are shown in light gray in the background. Residue bundles that are encouraged or discouraged in a concerted manner (as revealed by their high importance in the pooling hidden layer of the neural network, **Table 2**) are highlighted with thick lines.

additional set of encouraged contacts in the occluded state is located between the intracellular gating helices TM10b and TM4, which are required to drive formation of the occluded state from the inward-facing conformation (**Drew et al., 2021**; **Wisedchaisri et al., 2014**).

Interestingly, we observe the formation of less well-known interbundle contacts between TM1 and TM5, and between TM3 and TM6 when transitioning from an outward-open to an outward-occluded state, possibly modulating the conformational change of TM1. In transition between outward- and inward-facing states, additional contacts are also seen between TM2 and TM11, both at the extracellular region, in which case they are encouraged in the inward-facing states, and at the intracellular region, in which case they are encouraged in the outward-facing ones. These previously unreported contacts appear particularly important for transporter function as they are at the basis of the rocker-switch motion passing through the occluded state. Taken together, the state-specific coevolution analysis is able to replicate the main structural transitions expected from comparing individual static structures, as well uncovering new contacts to be evaluated (**Table 2**).

## Generating sugar porter structures for each major conformation

To expand the available structural information for sugar porters, AlphaFold2 models were driven into all five conformational states in RosettaMP (**Alford et al., 2015**) using the derived encouraged and

**Table 2.** The most important residue contacts as identified by the neural network of *Figure 3*. The residue numbering is relative to GLUT5. The contact bundles identified in *Figure 3* were extracted from the first pooling layer, such that 5 × 5 connected residue interaction matrices were tracked. Here, the top contacts of these matrices are listed according to their raw layer-wise relevance backpropagation (LRP) value for each state. A negative sign on the LRP coefficients signifies a contact that is absent in a certain state.

**Outward open**

| Contact | Helices involved | Contact bundle | LRP |
|---|---|---|---|
| Y297-N40 | TM7b–TM1 | 1 | −3.21 |
| Y298-V39 | TM7b–TM1 | 1 | −2.34 |
| F430-L70 | TM2–TM11 (top) | 3 | −2.31 |
| W71-F432 | TM2–TM11 (top) | 3 | −2.24 |
| P193-P112 | TM6–TM3 | 4 | −2.01 |
| A177-A38 | TM5–TM1 | 6 | −1.7 |
| F412-L87 | TM2–TM11 (bottom) | 3 | 2.20 |
| N157-E337 | TM5–TM8 | 2 | 2.48 |
| A167-V336 | TM5–TM8 | 2 | 2.76 |

**Outward occluded**

| Contact | Helices involved | Contact bundle | LRP |
|---|---|---|---|
| F430-L70 | TM2–TM11 (top) | 3 | −3.09 |
| W71-F432 | TM2–TM11 (top) | 3 | −2.97 |
| Y298-V39 | TM7b–TM1 | 1 | −1.31 |
| Y297-N40 | TM7b–TM1 | 1 | 1.21 |
| F412-L87 | TM2–TM11 (bottom) | 3 | 2.12 |
| A177-A38 | TM5–TM1 | 6 | 2.48 |
| P193-P112 | TM6–TM3 | 4 | 2.58 |
| N157-E337 | TM5–TM8 | 2 | 2.94 |
| A167-V336 | TM5–TM8 | 2 | 2.97 |

**Occluded**

| Contact | Helices involved | Contact bundle | LRP |
|---|---|---|---|
| P147-L397 | TM4–TM10b | 5 | 1.13 |
| A396-P147 | TM4–TM10b | 5 | 1.19 |
| T400-P147 | TM4–TM10b | 5 | 1.43 |
| R159-E401 | TM4–TM10b | 5 | 2.19 |
| F412-L87 | TM2–TM11 (bottom) | 3 | 2.25 |
| F430-L70 | TM2–TM11 (top) | 3 | 2.46 |
| W71-F432 | TM2–TM11 (top) | 3 | 2.50 |
| N157-E337 | TM5–TM8 | 2 | 2.94 |
| A167-V336 | TM5–TM8 | 2 | 2.97 |
| Y298-V39 | TM7b–TM1 | 1 | 3.06 |
| Y297-N40 | TM7b–TM1 | 1 | 3.98 |

*Table 2 continued on next page*

*Table 2 continued*

**Inward occluded**

| Contact | Helices involved | Contact bundle | LRP |
| --- | --- | --- | --- |
| P147-L397 | TM4–TM10b | 5 | −2.38 |
| A396-P147 | TM4–TM10b | 5 | −2.17 |
| T400-P147 | TM4–TM10b | 5 | −2.16 |
| F412-L87 | TM2–TM11 (bottom) | 3 | −1.96 |
| N157-E337 | TM5–TM8 | 2 | 1.23 |
| A167-V336 | TM5–TM8 | 2 | 1.48 |
| F430-L70 | TM2–TM11 (top) | 3 | 2.97 |
| W71-F432 | TM2–TM11 (top) | 3 | 2.98 |
| R159-E401 | TM4–TM10b | 5 | 3.15 |
| Y298-V39 | TM7b–TM1 | 1 | 3.25 |
| Y297-N40 | TM7b–TM1 | 1 | 3.27 |

**Inward open**

| Contact | Helices involved | Contact bundle | LRP |
| --- | --- | --- | --- |
| P147-L397 | TM4–TM10b | 5 | −3.12 |
| R159-E401 | TM4–TM10b | 5 | −3.01 |
| F412-L87 | TM2–TM11 (bottom) | 3 | −2.36 |
| N157-E337 | TM5–TM8 | 2 | −2.30 |
| A167-V336 | TM5–TM8 | 2 | −2.15 |
| A396-P147 | TM4–TM10b | 5 | −1.30 |
| T400-P147 | TM4–TM10b | 5 | −1.05 |
| F430-L70 | TM2–TM11 (top) | 3 | 2.34 |
| W71-F432 | TM2–TM11 (top) | 3 | 2.38 |
| Y298-V39 | TM7b–TM1 | 1 | 2.85 |
| Y297-N40 | TM7b–TM1 | 1 | 2.92 |

discouraged state-dependent contacts as attractive and repulsive biases, respectively (*Figure 2D*, Methods). A straightforward assessment of the quality of the models obtained was difficult to perform, however, as few experimental structures are available. Nevertheless, for an overall approximation, we calculated the Cα root mean squared deviation (RMSD) to the phylogenetically closest available experimental structure. As shown in *Figure 4*, the mean RMSD difference has an acceptable value of ~2.5 Å, with the RMSD distribution spanning a 95% confidence interval of 1.5–3.2 Å. In addition, to evaluate the stability of these models, 10 ns MD simulations of these models embedded in a model membrane bilayer were then performed, which further relaxed the structures, bringing their overall RMSD to the closest homolog to ~2.1 Å with the RMSD distribution spanning a 95% confidence interval of 1.1–2.7 Å. The RosettaMP modeling pipeline thus seems to bring the models close to their local free-energy minimum, but relaxation by MD simulations further improves the quality of the models by relaxing them further. To evaluate the cause of the RMSD reduction, we compared the GLUT5 outward-open state model before and after MD relaxation to the crystal structure of the same transporter solved in the same state (PDB ID 4YBQ), which showed that the model quality was improved from a 2.15- to a 1.49-Å Cα RMSD, mainly via the relaxation of TM1 (*Figure 4C*). It seems that, in this case, flexible helices require an explicit solvent model to attain a more physiological conformation.

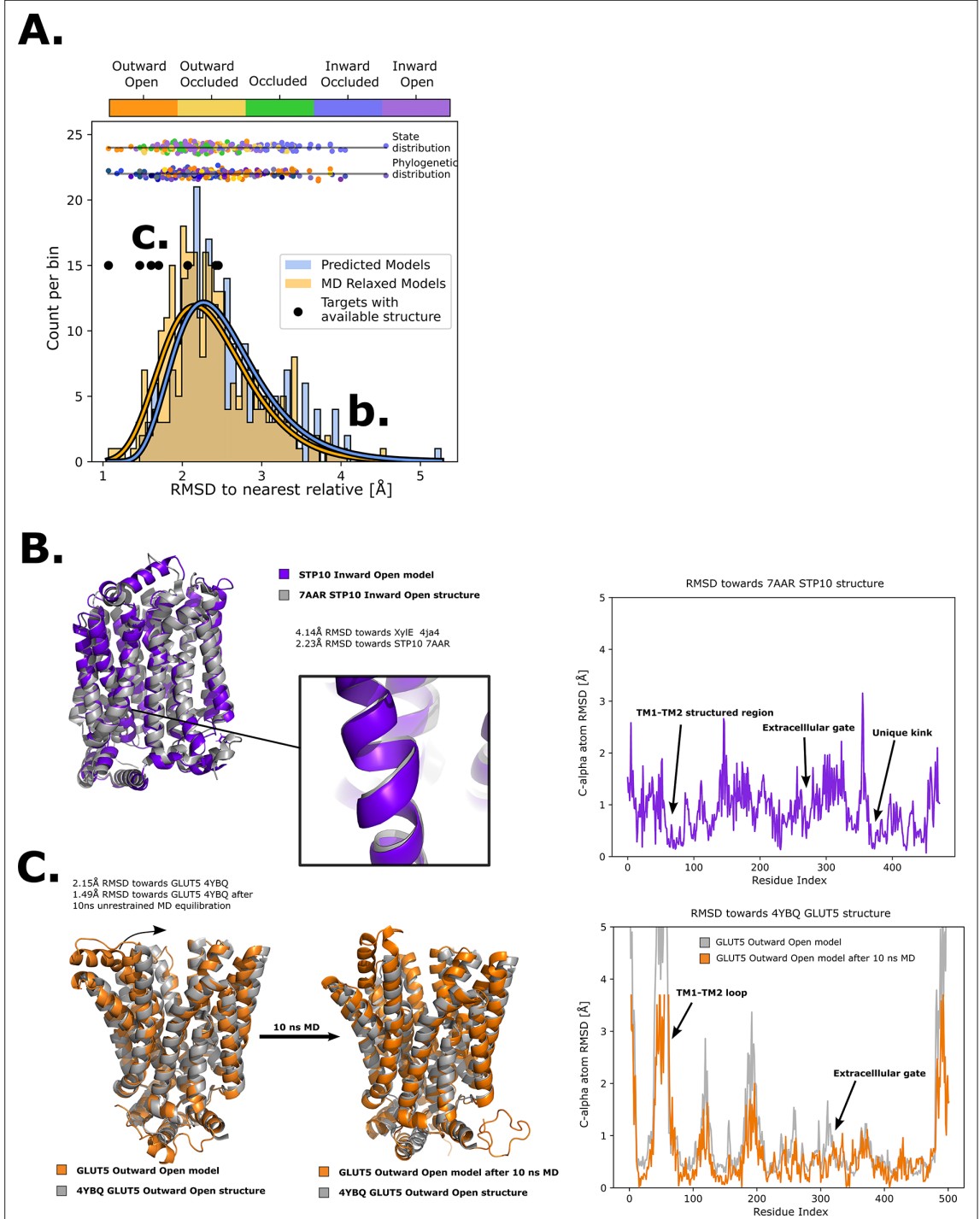

**Figure 4.** Quality of structure prediction of state-specific models of SPs. (**A**) Histogram of root mean squared deviation (RMSD) of the structural models (Rosetta, blue and molecular dynamics [MD] relaxed, orange) from their closest relative in the same conformational state. The distributions of the MD-relaxed structures colored according to state and phylogeny (see color definition in *Figure 1C*) are shown above the histogram. Additionally, the targets with available experimental structures are indicated with black dots. (**B**) Alignment between STP10 inward open model and the newly solved 7AAR STP10 structure in the same conformational state. A helical kink not present in any experimentally determined structure so far is shown as an example of a transporter-specific feature that is captured by our structure prediction protocol. The residue-wise RMSD between our model and the experimentally determined structures is shown to the right, where the TM1–TM2 loop, extracellular gate and the aforementioned unique kink are highlighted. (**C**) (left) Alignment between the GLUT5 outward-open model and an experimental structure of the same isoform in the same state (PDB ID 4YBQ). (right) Alignment between the MD-relaxed GLUT5 outward-open model and PDB 4YBQ. The residue-wise RMSD between the MD models and the experimental structure is shown to the right, where the regions improved after 10 ns MD simulations are highlighted and labeled.

As a next assessment of quality, we re-analyzed the STP10 model against the inward-open STP10 crystal structure (PDB ID 7AAR) (*Bavnhøj et al., 2021*), whose structure was not present in our initial training set as its structure was released after training of our model. The inward-open state model of STP10 previously had a calculated Cα RMSD of 4.3 Å away from its closest homologue with known structure, XylE (PDB 4QIQ) (*Wisedchaisri et al., 2014*). However, the actual Cα RMSD to the experimental STP10 inward-facing structure difference is only 2.2 Å, which was further reduced to 1.8 Å after MD simulations. Inspection of the STP10 structure reveals a unique helical kink in TM10, which is not present in XylE (*Figure 4B*) nor in the starting structure of STP10 in the outward-occluded state (PDB ID: 6H7D) (*Bavnhøj et al., 2021*). An additional structure for inward-open GLUT4 (*Yuan et al., 2022*) was further determined after our initial model training was finalized. Our inward-open GLUT4 model has a RMSD of 2.3 and 1.8 Å before and after MD relaxation, respectively. Reassuringly, our method is thus able to model new structural features absent from the training set. Indeed, the Cα RMSD estimated for HXT and STP members, as well as for models of the inward-occluded states, which have fewer structures available to properly assess their accuracy, is generally higher (*Figure 4A*). Overall, we conclude that the generated sugar porter models are of sufficient quality for linking conformational states of sugar porters.

## A free-energy landscape for the fructose transporter GLUT5

Characterizing robust free-energy landscapes of conformational cycles of SPs will enable us to better understand the mechanistic basis for sugar transport. Although free-energy landscapes for GLUT transporters have been reported previously (*Ke et al., 2017*; *Park and Huang, 2015*; *Galochkina et al., 2019*; *Chen and Phelix, 2019*), these were generated from a few structures only, and lacked the most recent structure in the occluded conformation (*Drew et al., 2021*; *Qureshi et al., 2020*). Indeed, as coevolution analysis confirms, the occluded state is an intermediate that has a number of important and specific coevolved pairs that are critical for linking the outward and inward-facing conformations. In addition, the methods used to generate these landscapes did not consider whether structural features captured in specific structures were transferable to other family members. Here, we decided to generate a free-energy landscape for the human fructose transporter GLUT5 using family-wide information in the form of coevolving contacts. We chose GLUT5 because it is the only transporter with experimental structures in both fully outward- and inward-facing conformations (*Nomura et al., 2015*) and because our recently determined free-energy landscape using a more traditional enhanced sampling method can serve as a comparison (*Chen and Phelix, 2019*).

The most efficient enhanced sampling techniques require the choice of a low-dimensional CV set that encompass all degrees of freedom implicated in a conformational transition (*Harpole and Delemotte, 2018*; *Lindahl et al., 2018*). Given that state-dependent contacts constitute the driving force for a conformational transition, CVs based on those are well suited to distinguish conformational states and to enhance transitions between them. We thus trained a support vector machine (SVM) to distinguish the modeled structures of adjacent states (*Figure 5—figure supplement 1*) and extracted therefrom top coevolving contacts, as inferred from their high SVM coefficients. We then designed state-specific CVs as weighted sums of distances between top state-specific coevolving pairs, using as weights the SVM coefficients (see Methods). Finally, we ran accelerated weighted histogram (AWH), an efficient enhanced sampling method natively available in GROMACS, using these CVs as input. Accumulating 250–650 ns of MD simulations enabled an extensive sampling of the conformational space and the estimation of the corresponding free-energy landscapes (*Figure 5A–C*) with satisfactory accuracy (*Figure 5—figure supplements 1 and 4* and *Figure 5—figure supplement 5*). The three separate free-energy landscapes give insights into the energetical details of each process. However, to infer global characteristics of the entire conformational cycle, we needed to estimate the free energy along a single variable able to describe the entire conformational cycle. To do this, we optimized a modified linear discriminant analysis (LDA) loss function to maximize the separation between states (see Methods). We then reweighted all of the samples from the separate AWH simulations, aggregating all of the statistical weights in a one-dimensional LDA-based CV space (*Figure 5D*). We projected the experimentally solved structures in this same CV space to help identify the free-energy basins and analyzed representative snapshots extracted from each basin (*Figure 5E*, *Figure 5—figure supplement 6*). As expected, the three AWH simulations mostly agree in the regions where they overlap, for example in the transition state region or relative free energies of the outward-occluded

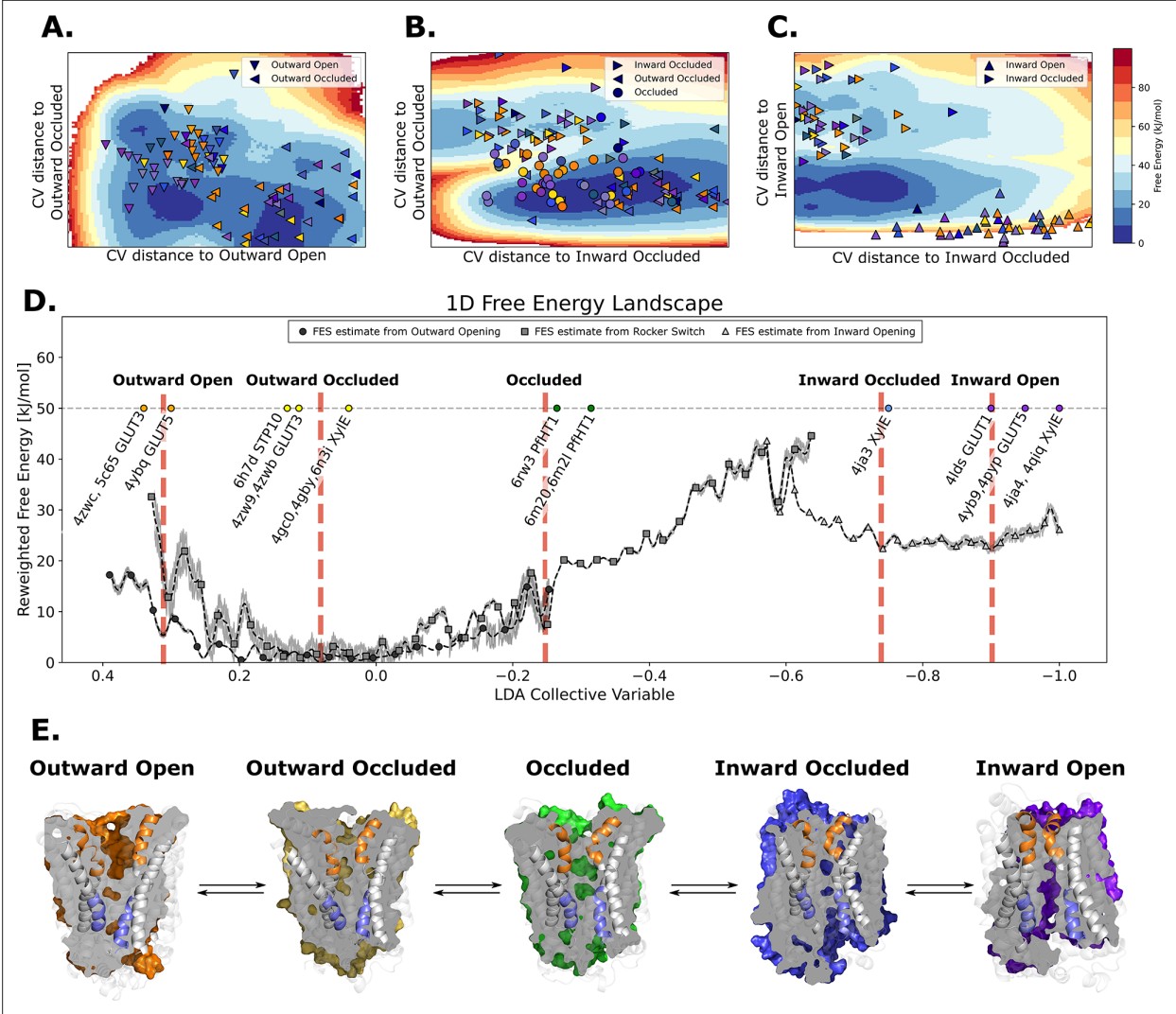

**Figure 5.** The entire conformational cycle is captured by accelerated weight histogram simulations. (**A**) Free-energy landscape of the outward opening process. The most populated free-energy basins are labeled according to annotation based on visual inspection and root mean squared deviation (RMSD) calculations of snapshots from the basins to the available experimentally determined structures. Projections of models of the outward-open and outward-occluded models of the various SPs are shown as symbols colored according to *Figure 1D*. Note that many outward-occluded models fall in the basin representing the occluded state, most likely because of the small structural difference between a bent and broken TM7b helix. (**B**) Free-energy landscape of the rocker-switch process. Representation as in A. Note that although not trained on the occluded models, models mostly either fall in the barrier region or in the GLUT5 occluded basin. (**C**) Free-energy landscape of the inward opening process. Representation as in A. (**D**) 1D free-energy landscape of the full conformational change, determined by reweighting the local free-energy estimates of panels A–C (labeled with different shapes) on a globally trained discriminative collective variable (see methods). The estimated point-wise error of the free-energy estimate is shown in gray. The experimentally solved structures were projected on the collective variable, and the lowest energy basin closest to the mean of each conformational state was extracted and analyzed and shown in panel E. (**E**) Representative structural models extracted from each of the most populated basins for each labeled state in panel D.

The online version of this article includes the following figure supplement(s) for figure 5:

**Figure supplement 1.** Convergence of accelerated weighted histogram (AWH) calculations.

**Figure supplement 2.** Collective variable learning.

**Figure supplement 3.** GLUT5, STP10, PfHT1, XylE, and HXT9 models projected on free-energy surfaces.

**Figure supplement 4.** Overlap and probability distributions.

**Figure supplement 5.** Transition imbalances and error estimation.

**Figure supplement 6.** Observable distributions.

**Figure supplement 7.** Predicted structure stability analysis.

and occluded states. Notably, the rocker-switch simulations covered the conformational space most extensively, briefly sampling even outward-open states. However, the free-energy uncertainty estimated from these simulations in the outward-open region is higher than in the rest of the conformational ensemble (~4–6 kJ/mol), most likely due to the fact that the rocker-switch CV was not explicitly trained to sample the outward-open state. In contrast, the free-energy estimate coming directly from the outward opening simulations is more reliable.

In the absence of substrate, the most favorable conformation for GLUT5 is the outward-facing state, which is consistent with biochemical analysis and the salt-bridge network on the intracellular side stabilizing this state (*Deng et al., 2015*; *Nomura et al., 2015*; *Schürmann et al., 1997*). The outward-open and outward-occluded states appear to be of comparable free energies, and the barriers between the outward-open and outward-occluded states are fairly low at 4.6–9.9 kJ/mol, respectively (*Figure 5E*). This observation is consistent with the fact that the structure of GLUT3 has been determined in both outward-open and outward-occluded states even in the presence of a bound maltose (*Deng et al., 2015*). The largest energetic barrier of ~35.7 kJ/mol is located between the occluded and the inward-occluded state (*Figure 5D*). This barrier presumably arises in part from the breakage of the strictly conserved salt-bridge network, is consistent with the activation barrier of 10 kcal/mol as estimated by GLUT1 kinetics (43 kJ/mol) (*Lowe and Walmsley, 1986*; *Ezaki and Kono, 1982*; *Figure 6E*). Notably, the relaxed GLUT5 occluded state does not fall on the largest energetic barrier corresponding to the transition state (*Figure 5*). However, in the presence of a substrate sugar, GLUT5 passes through a transition state that closely matches the occluded state of *Pf*HT1 (*McComas et al., 2022*). Our analysis indicates that coevolution-driven MD simulations are detecting an energetic minimum for an occluded state prior to the transition state which has not yet been experimentally observed (*Figure 5—figure supplement 3*). It is possible that the coevolution analysis is detecting a pre-transition state, which is likely to be lowly populated in the presence of a substrate sugar. As is expected in the absence of substrate, we fail to entirely stabilize the inward open state, which is also evident from the disparity in the localization of the inward open models and the corresponding free-energy basin (*Figure 5—figure supplement 6*). Taken together, we conclude that using evolutionary-based CVs enables us to obtain sugar porter conformational free-energy landscapes using AWH with simulations of the order of hundreds of nanoseconds. While not strictly comparable to other methods, this represents an increase in performance by at least an order of magnitude compared to some previous attempts on modeling conformational change of comparable magnitude (*Galochkina et al., 2019*; *Selvam et al., 2018*; *Takemoto et al., 2018*).

Since computationally predicted models are not guaranteed to be stable over time in MD simulations, we evaluated the stability of the predicted models of GLUT5 by performing 500 ns equilibrium MD simulations of each of the conformational states used in AWH (*Figure 5—figure supplement 7*). In these simulations, we observed instability of the outward- and inward-open models, which we could attribute to occlusion of the extra- and intracellular gates (*Figure 5—figure supplement 7A*). Reassuringly, conformational changes at this scale are expected based on the free-energy landscape we obtained (*Figure 5D, E*).

## Family-wide state-dependent interactions

Having validated that the top coevolving pairs can be combined into CVs able to connect the different conformational states and describe the energetics of transitions between them appropriately, we scrutinize the interactions that went into the construction of the process-specific CVs by projecting them onto the models of GLUT5 states (*Figure 6*). Notably, these interactions are more robustly defined than in the previous analysis shown in *Figure 3*, since those were based on the analysis of a reduced set of experimentally resolved structures. In addition, we track the functional diversity across the SP family by extracting the types of residue interactions found at these sites (*Figure 6*).

Starting from the outward-open state, as expected, salt-bridge forming residues appear to stabilize the outward-facing conformations, a network that is maintained through to the occluded state (*Figure 6A*). In addition, the residue Ser306 located at the back of TM7b has coevolved to form an interaction to Gln366 located in TM9, an interaction that is between Thr/Arg or Lys in the SP members or made between small and/or polar residues in the rest of the family (*Figure 6B*).

The outward-occluded state is characterized by a bend in the TM7b helix, which occludes sugar exit (*Drew et al., 2021*; *Sun et al., 2012*). The contact analysis shows that this state appears to be

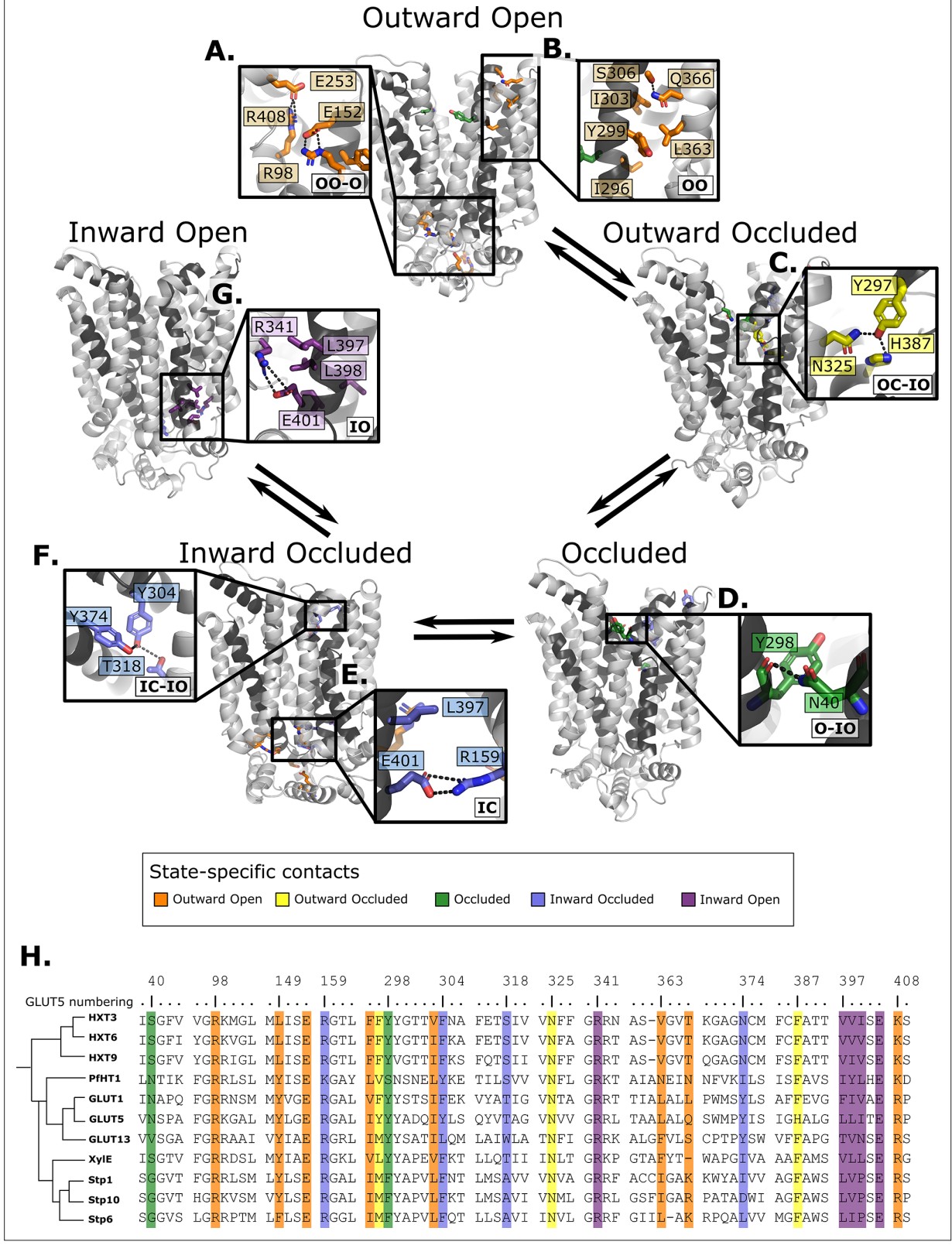

**Figure 6.** Summary of the structural determinants responsible for the cycling between adjacent conformational states. Contacts are characteristic of the first conformation they appear in, but they can be maintained more or less throughout the cycle depending on the family members. The snapshots are of the GLUT5 models used as a representative member. The bottom right of every panel contains information about which states the contacts are present in (outward open/occluded [OO/OC], occluded [O], or inward open/occluded [IO/IC]). (**A**) Salt-bridge network at the intracellular side.

*Figure 6 continued on next page*

*Figure 6 continued*

This network is intact in all outward-facing states, but breaks during the rocker-switch motion (except for the E401/R159 contact, see panel E). (**B**) TM7b–TM9 interface responsible for stabilizing the outward-open conformation of TM7b, of which the hydrophobic contacts are only present in the outward-open state. (**C**) TM7b contact network responsible for promoting both the bent and broken conformation of TM7b. The position of N325 and H387 necessitates the rotation of TM7b which occludes the extracellular gate. (**D**) Backbone contact formed between TM7b and TM1, which is only possible when TM7b is completely broken. (**E**) TM10b–TM4 interactions that are the last occluding contacts to break before the intracellular gate opens. (**F**) TM10a–TM8 contacts responsible for stabilizing the new interbundle angle present in the inward-facing states. (**G**) Salt bridge and hydrophobic nexus responsible for stabilizing the inward-open conformation of TM10b, which fully unblocks the binding site from the intracellular side. (**H**) Multiple sequence alignment (MSA) of some representative members at positions shown throughout panels A–G. Since model training was performed on all predicted models using highly coevolving residue pairs as features, the type of interaction present in different subfamilies can be tracked in this MSA.

The online version of this article includes the following figure supplement(s) for figure 6:

**Figure supplement 1.** Intracellular gate evolution.

stabilized by coevolving pairs between Tyr297 in TM7b and Asn325 and His387 in TM10a, which are peripheral to the sugar-binding site and could therefore be connected to sugar binding (*Figure 6C*). This interaction remains formed up until the inward occluded-open state. In most other GLUT members and in the HXT proteins, Tyr297 and His387 are replaced by Phe residues that are expected to interact with TM10a residues via hydrophobic or pi–pi interactions.

In the occluded state, TM7b transitions from a bent to a broken helix, which occludes the sugar-binding site and moves TM7b closer to TM1. Interestingly, Asn40 of TM1 forms a hydrogen bond with the backbone residues at the break-point in TM7b, an interaction that remains formed up until the inward open states (*Figure 6D*). The Asn40 is conserved in *Pf*HT1 and mutation to alanine was shown to severely impair transport (*Qureshi et al., 2020*). In most other family members, a hydrogen bond is made possible by the presence of a small polar residue in TM1, highlighting that stabilized TM7b closure is generally connected to interactions in TM1 as early as in the occluded state.

In addition to the most outward-facing salt-bridge network (*Figure 6A*) breaking, rearrangements of TM10a helix and formation of the Tyr374 (TM10)–Thr318 (TM8) hydrogen bond characterizes the transition to the inward-facing states (*Figure 6F*). These positions often feature residue pairs capable of interaction either via hydrophobic interactions or H-bonding (Leu/Val-Ala in STPs, Asn/Tyr-Thr/Ser in the rest of the family). The formation of this contact in the inward-occluded state is a critical determinant for the rocker-switch motion, as the resulting rearrangement of TM10a promotes the tilting of the peripheral helices, which ultimately rocks the helical bundles into an inward-facing state.

For the intracellular gate to open, the residual Glu401–Arg159 salt bridge (*Figure 6E*) must completely break. Inspection of the distribution of the minimum distance between Glu401–Arg159 contact across reveals that it increases from an average of 1.4 Å in the occluded state, to 3.4 Å in the inward-occluded state. The weakening of Glu401–Arg159 presumably contributes to releasing TM10b. Entry into the inward-open state in the absence of sugar, stabilizing the tilted TM10b, features the full breakage of the Glu401–Arg159 interaction, and Glu401 forming a salt bridge with a different partner, namely Arg341 (*Figure 6G*). The rotation of TM10b shifts the occluding Leu397 residue away from intracellular pathway to the sugar-binding site, opening the intracellular gate fully. This state is also stabilized by additional strictly conserved hydrophobic contacts in the TM10–TM7 interface (positions 397–400 in GLUT5, which rearrange from an interaction with TM4 in the outward-facing and occluded state) (*Figure 6G* and *Figure 6—figure supplement 1*).

## Coevolving residues support proton-coupling in sugar porters

Having characterized the aspects of the sugar porter transport cycle that are likely conserved across the family, we turn to family-specific features, with a focus on the differences between passive and proton-coupled transporters, such as XylE. The current working model is that an aspartic acid reside in TM1 (Asp27 in XylE) is allosterically coupled to the sugar transport but does not participate directly in sugar binding (*Drew et al., 2021*; *Ke et al., 2017*; *Seica et al., 2020*; *Jia et al., 2020*). Indeed, the aspartic acid to asparagine mutant has the same sugar-binding affinity as wild type (*Madej et al., 2014*). Rather, it is thought that the protonation of the TM1 aspartic acid is required for transport as it breaks the outward-facing-specific salt-bridge interaction to an arginine residue Arg133 on TM4. Based on the occluded *Pf*HT1 structure (*Drew et al., 2021*), highlighting the coupling between TM1 and TM7b, a latch mechanism for proton coupling was proposed (*Drew et al., 2021*). Simplistically,

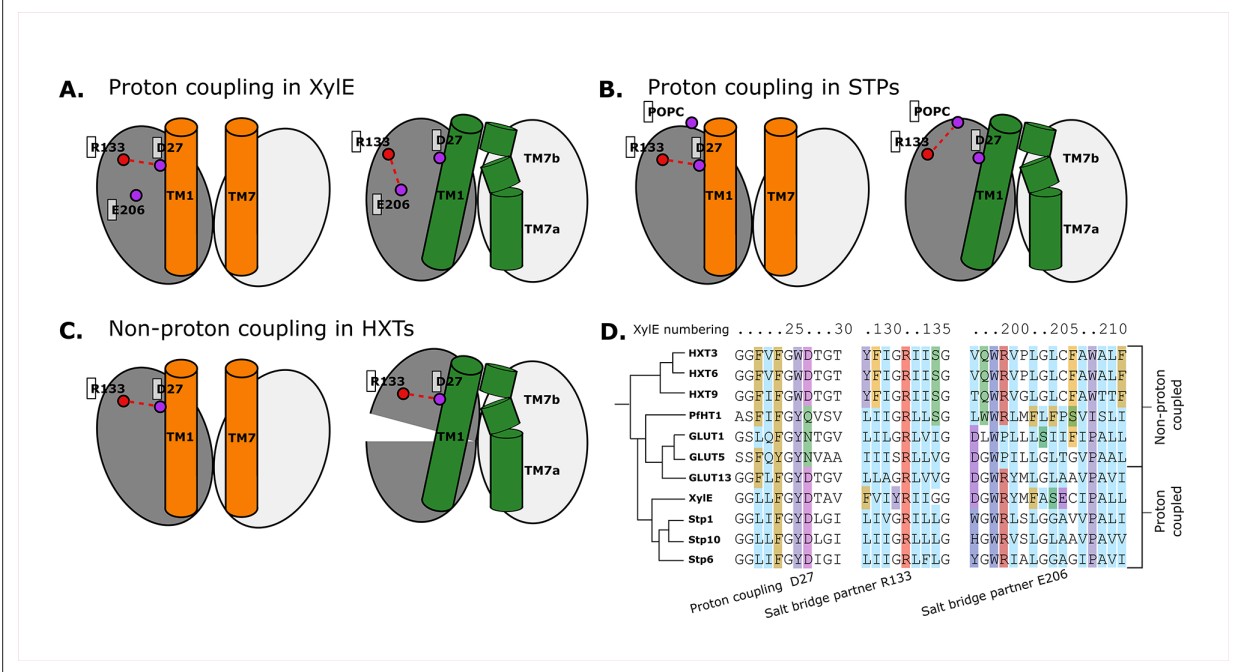

**Figure 7.** Structural and evolutionary basis for proton coupling in the sugar transporter (SP) family. (**A**) Proposed cartoon model for proton coupling in XylE, showing the highlighted salt-bridge network rearrangements between the outward-open (orange) and occluded (green) conformational states. Note how these rearrangements facilitate the closure of the extracellular gate. (**B**) Proposed cartoon model for proton coupling in the SPs, showing the highlighted salt-bridge network rearrangements between the outward-open (orange) and occluded (green) conformational states. Note how these rearrangements facilitate the closure of the extracellular gate, and how R133 interacts with another latent salt-bridge partner than in XylE. (**C**) Proposed cartoon model explaining why HXTs generally lack proton coupling, showing the highlighted salt-bridge network rearrangements between the outward-open (orange) and occluded (green) conformational states. Note how the crucial R133–D27 contact can stay formed even in the occluded state due to a tilt of the entire helical bundle. (**D**) Multiple sequence alignment (MSA) of a selected subset of proton- and non-proton-coupled SPs, with residues labeled according to the XylE numbering.

The online version of this article includes the following figure supplement(s) for figure 7:

**Figure supplement 1.** Structural and evolutionary basis for proton coupling in the sugar transporter family with an atomic resolution.

breakage of the Asp27–Arg133 salt bridge enables TM1 to come closer to TM7b which is associated with sugar binding and closing of the extracellular gate. Formation of the occluded state is catalyzed by sugar binding, but in proton-coupled transporters, this is only enabled by the release of the latch (***Figure 7A*** , ***Figure 7—figure supplement 1A***).

The coevolution analysis is consistent with TM1 and TM7 interactions driving formation of the occluded state. More specifically, we indeed see that the equivalent TM1 Asp and TM4 Arg salt bridge is broken in occluded states of all known proton-coupled symporters, such as STP10. The details of the conformational rearrangement differ, however. After breaking of the latch, in the occluded state of XylE, Arg133 can form a salt bridge with the neighboring TM6 Glu206 residue (***Figure 7A***). In STP10, on the other hand, Glu206 is replaced by an alanine, and R133 rearranges by adopting a different rotameric state facing toward TM11, where it appears to be stabilized by an interaction with the phospholipid headgroups (***Figure 7B*** , ***Figure 7—figure supplement 1B***).

Most GLUTs are thought to be passive transporters and do not possess an acidic residue at position 27, consistent with its purported role as proton carrier (***Figure 7D***). Intriguingly, however HXTs, which are generally not proton coupled, do carry an acidic residue in this position. Our analysis reveals that this is explained by the fact that these transporters contain a latch interaction that is maintained throughout the conformational cycle thanks to a large-scale pivot of the first helical bundle (***Figure 7C*** ***Figure 7—figure supplement 1C***). This conformational change unique to HXTs is enabled by helix breaking residues at position 202 (Pro/Gly), which are replaced by mostly hydrophobic residues in the rest of the family (***Figure 7D***). GLUT2, a passive transporter that nevertheless possesses an acidic

residue at position 27, has a serine in position 205 which interacts with D27, which appears to help the salt bridge remain formed in the occluded state (*Figure 7—figure supplement 1D*).

## Discussion

AlphaFold2 has made it possible to produce high-quality models of structures associated with any amino acid sequence (*Jumper et al., 2021*). Nevertheless, these structures represent an arbitrary conformational state, and do not capture the conformational heterogeneity that enables protein function. Modifying the MSA depth enables to make models of alternate states, though in a non-deterministic manner (*Del Alamo et al., 2022*). In this work, we have used (1) AlphaFold models, (2) coevolution analysis encoded by family-wide MSAs, and (3) state-specific contacts extracted from the ~20 structures to guide enhanced MD simulations. Our work goes beyond the analysis of static structures and enables the separation of state- and transporter-specific features, thus firmly establishing the family-wide determinants of the conformational cycle as well as establishing specificities of GLUT5 and elucidating the mechanism of proton-coupled SPs. By introducing coevolution as a measure of predicted interaction, we can develop models that go beyond the comparison of structures only. We also minimize information that may be lost during dimensionality reduction, as demonstrated by our validation on newly reported structures.

Using state-dependent coevolution-based contacts, we show that we can create a low-dimensional projection that describes the transition between adjacent states. When derived into a set of CVs used in enhanced sampling MD simulations, we can construct weighted conformational ensembles, or in other words free-energy landscapes of the process. Comparing our convergence times (to less than 0.01 kJ/mol within 250–650 ns per walker and system) to previous works on similar systems (*Galochkina et al., 2019*; *Selvam et al., 2018*; *Takemoto et al., 2018*) and given the agreement between the activation barrier estimates from kinetic measurements (*Lowe and Walmsley, 1986*; *Ezaki and Kono, 1982*), we conclude that the introduction of coevolution data is a powerful approach in obtaining accurate free-energy landscapes efficiently. This methodology is likely to be applicable to other membrane protein families, such as GPCRs, ABC transporters, or other SLC transporter families.

The conformational ensembles reveal an overall functional cycle that is largely consistent with the evaluation of experimentally resolved structures, yet also provides details that were previously unknown. Across the larger MFS family of transporters, cavity-closing contacts are predominantly formed between TM1–TM2 and TM7–TM8 on the outside and between TM4–TM5 and TM10–TM11 on the inside (*Drew et al., 2021*; *Drew and Boudker, 2016*). However, the sugar porters, as exemplified by GLUT transporters, coordinate the sugars very asymmetrically with only a single residue in the N-terminal bundle coordinating the sugar. As such, local rearrangements during the transport cycle are assumed to be primarily established by local changes in TM7b and TM10b half-helices, which coordinate the sugar in the C-terminal bundle. Although sugar porters are made up of two structurally similar bundles, the asymmetrical rearrangements are more akin to the conformational changes described by rocking-bundle proteins (*Drew et al., 2021*; *Drew and Boudker, 2016*), which are made up from structurally distinct bundles. Consistently, coevolution analysis is able to detect the importance of the substrate gating regions for driving conformational changes. In particular, the coevolution analysis shows that TM7b and its interaction with TM1 have evolved to come together already in the occluded state, that is, rather than only interacting in the inward-facing conformations. This conclusion is in agreement with the occluded structure seen for *Pf*HT1 (*Qureshi et al., 2020*), wherein TM7b had moved completely inwards to break over the sugar-binding site.

The requirement for TM1 to interact with TM7b in formation of the occluded state further explains how sugar porters can evolve to be proton coupled, even if the proton-accepting residue itself does not directly coordinate the sugar, that is, unlike in other SPs, such as LacY. Simplistically, an aspartic acid residue in TM1 forms a salt bridge with an arginine residue in TM4, restricting its movement. Protonation of the aspartic acid residue removes this constraint and TM1 can come closer to TM7b, which itself is stabilized inwards when it binds a substrate sugar. Interestingly, an acidic residue in TM6 is further required to provide a favorable alternative conformation for the unpaired arginine. In the passive GLUT transporter GLUT2 and the yeast hexose transporters, for example, this additional acidic residue is missing and the salt bridge is still retained due to the assistance of a nearby serine residue. Nevertheless, STP members are proton coupled and yet they do not possess an additional acidic residue in TM6. Rather, the arginine residue is able to snorkel to the membrane interface in the

unpaired state. Notably, although TM7b and TM1 are important, they are part of a larger interaction network that also includes, for example, how TM7b interacts with TM8 in the outward-open state. Further analysis and simulations are required, however, to identify the allosteric network responsible for proton transfer.

Consistent with the orchestrating role of TM7b in substrate translocation, the coevolved interaction with TM1 is retained across the entire second half of the transport cycle. In contrast, the inward-facing gating helix TM10b is likely to have a more passive role and a coevolved interaction with TM4 is only formed in the occluded state. Nevertheless, TM10b dynamics are likely to be important and it is possible that TM10b movement is facilitated by the stabilization of TM10a with TM8 on the extracellular side (*Figure 6F*). The bottom of TM10a harbors a strictly conserved acidic residue, which is part of the intracellular salt-bridge network. The contact maps initially generated for all sugar porters indicated mostly discouraged contacts between TM4 and TM10b residues in the inward-occluded state (bundle 5 contact in *Figure 3*). In addition, the GLUT5 models show that whilst most contacts are indeed broken, there is still an encouraged contact between TM4 and TM10b residues in the inward-occluded state (*Figure 6F*, *Figure 6—figure supplement 1B*). It thus appears that, at least in GLUT5, only one of the salt-bridge pairs is fully broken between the occluded and inward-occluded states. Such a result is intriguing and implies that even the rocker-switch rearrangement itself, might utilize asymmetric rearrangements, contrary to the symmetric bundle movement in other members of the MFS family, such as SWEETs (*Drew et al., 2021*).

Lastly, GLUT transporters are shown as textbook examples of how small molecule transporters are functional equivalents to enzymes (*Drew et al., 2021*). Key to understanding catalysis is to understand how the transition state is formed during substrate translocation. The transition state, however, is only transiently occupied and therefore difficult to experimentally capture. Nevertheless, the parasite transporter *Pf*HT1 has an unusually very polar TM7b gate, which has made it possible to capture an occluded state with sugar present (*Qureshi et al., 2020*; *Jiang et al., 2020*) that is, notably this state is unlikely to represent a transition state for *Pf*HT1. Nonetheless, out of the models we built, *Pf*HT1 has the highest number of state-specific contacts in both gates. We speculate that the *Pf*HT1 model could represent a pre-transition state (*Figure 5D*), with further dynamics between the bundles required to access the transition state. In summary, although many questions remain unanswered, our novel approach provides a rational framework for understanding how sugar porters function at the molecular level and provides the information to re-engineer sugar porters with different characteristics, which might be otherwise inaccessible by traditional forward-evolution approaches. Through the identification of family-wide state-specific interaction patterns, it would be possible to guide experimental dynamics, such as single-molecule FRET or $^{19}$F NMR experiments. We also anticipate that breaking or stabilizing the interaction patterns would perturb the free-energy landscape significantly since these contacts are tightly coupled to survival through evolution. At a conceptual level, our work highlights how even the most simplistic type of transporters have evolved fine-tuned and intricate interactions to achieve substrate translocation.

## Methods

### Sequence library construction

To construct the sequence alignment at the basis of coevolution analysis, a representative sequence from each member of the SP family was taken as a seed for a sequence search against the Uniref90 database (*Suzek et al., 2015*) using PSIBLAST (*Altschul et al., 1997*). In this way, a sufficient number of diverse sequences was consistently found around each member ($M_{eff} > 1000$, where $M_{eff}$ denotes the number of sequences with less than 80% identity). Because of the local searches in sequence space, we did not manually trim the sequences or exclude any sequences. Additionally, for the coevolution analysis we automatically filtered out positions with more than 20% gaps. The sequence libraries were then aligned within each subfamily using a stair-shaped guide tree (*Boyce et al., 2014*) in the MUSCLE alignment algorithm (*Boyce et al., 2014*; *Edgar, 2004*) with the default parameters. The sequence alignments and final MSA used for the coevolution analysis can be found in the Zenodo repository.

## Coevolution analysis

The resulting MSAs were then used as input for DCA, where the aim is to fit a Potts model with functional form *Morcos et al., 2011*:

$$P\left(x_i^n|x_{-i}^n\right) = \frac{1}{Z_i}\exp\left(v_i\left(x_i^n\right) + \sum_{j=1}^{L} w_{i,j}\left(x_i^n, x_j^n\right)\right) \tag{1}$$

where the conditional probability represents the information from the entire MSA given the model parameters $v_i$ (representing position-wise information) and $w_{i,j}$ (representing pair-wise information). The notation $x_{-i}^n$ denotes the MSA without accounting for the *i*th sequence. Thus, for every sequence, all other sequences are used to estimate the probability of observing the parameters. Parameters were fitted using maximizing the pseudolikelihood of observing a set of *N* sequences of length *L* (*Kamisetty et al., 2013*):

$$L\left(v, w| \left(x_0^0, \ldots, x_L^0\right) \cdots \left(x_0^N, \ldots, x_L^N\right)\right) = \sum_{n=1}^{N} \sum_{l=0}^{L} \ln P\left(x_i^n|x_{-i}^n, v, w\right) \tag{2}$$

Since the information about the pair-wise information is contained within the w parameter, the evolutionary couplings $C_{i,j}$ in a coevolution matrix *C* were calculated according to *Equation 3*, and afterwards standardized to the $N\left(0, 1\right)$ distribution.

$$C_{i,j} = \sum_{A=0}^{20} \sum_{B=0}^{20} w_{i,j}\left(A, B\right) \tag{3}$$

The resulting coevolution maps had an average false positive rate of 8.4% for the top 500 coevolving pairs, which corresponds to an average of 42 contacts. The coevolution maps from each sequence cluster were combined into a global coevolution map by calculating the average coevolution score at each position. Since members of the family were picked roughly equidistantly in sequence space (a sequence identity of around 50%, where possible), coevolution maps were weighted equally. In general, this procedure ensures that family-wide features are upweighted, whereas features present in only a few coevolution maps are down-weighted.

Importantly, we did not choose to include all contacts, but merely the top *t* ones that were determined to be sufficient for distinguishing conformational states. *t* was determined using a procedure based on the available experimental structures. First, experimental contact maps were calculated, where contacts were estimated from distances between pairs of residues filtered by a sigmoid function which introduces a smoothened cutoff at 4 Å:

$$M\left(A, B\right) = \frac{1}{1+\exp\left(2r_{A,B} - 8\right)} \tag{4}$$

where $r_{A,B}$ is the minimum distance between residues *A* and *B*.
Second, similarity between contact maps was defined as

$$d\left(x, y\right) = \langle C_{tot} \otimes M_x, C_{tot} \otimes M_y \rangle \tag{5}$$

which was used to calculate a distance matrix between all experimental structures. Third, UMAP (*McInnes et al., 2018*) was used to obtain a 1D embedding of all experimental structures from the distance matrix, for which the pair-wise symmetrical KL-divergence (*Kullback and Leibler, 1951*) between the five distributions was calculated. This KL-divergence was minimized with respect to the parameter *t*. The described procedure yielded a *t* of 183.

## State-specific structure prediction

A CNN was trained to classify the five functional states using contact maps from experimentally determined structures.

The input layer was of the size $N^2 \times C$, where *N* is the number of total number of MSA columns, and *C* the number of experimentally determined contact maps. The output layer was a simple vector with five nodes, each corresponding to an experimentally characterized state.

The network architecture was designed to filter experimental contact maps by coevolution scores, and to avoid redundancy between adjacent residues involved in contacts. To implement these criteria, the input layer was followed by a filtering layer based on the optimized coevolution maps and by a

pooling layer that gathered contacts formed by adjacent residue pairs. Lastly, two convolutions with sigmoid activation were applied to yield the output layer through an intermediate hidden layer. The 30-dimensional hidden layer was included to ensure that one contact could appear combinatorically in different states.

The loss function (*Equation 6*) contained *L2* regularization to prevent explosion of the weights:

$$L\left(e, d\right) = \sum_{i=0}^{n} \left(e_i - d_i\right)^2 + \sum_{i=0}^{n} e_i^2 \tag{6}$$

where *e* is the value of the output nodes during training and *d* the target values of the same nodes. Training involved thorough regularization and constriction for example by the aforementioned pooling. However, since the training set only contained contact maps from 36 individual chains, many of which originating from the same deposition, overfitting the network was unavoidable (*Hawkins, 2004*). The resulting model should thus not be used to make predictions.

To identify encouraged and discouraged contacts for each state, we performed LRP (*Bach et al., 2015*) on all five output classes separately. Moreover, these scores are visualized for each feature contact pair with >0.1 relevance in *Figure 3*, where highly scoring pairs in the pooling layer are high-lighted with thick bands.

The contacts that scored more than 0.1 through LRP were used to apply attractive and repulsive biases to guide starting models toward the desired functional state. Specifically, initial structures for each family member were downloaded from the AlphaFold2 database website. Then, Multi Constraint (repulsive) and Ambiguous Constraint (attractive) bias functions were applied to all heavy-atom distance pairs in the RosettaMP minimization scheme with implicit solvent and membrane (*Alford et al., 2015*). For repulsive Multi Constraint, the built-in Rosetta fade function was applied. For attractive Ambiguous Constraint, simple flat harmonics were applied. The force constants were deliberately chosen as weak (<20 Rosetta standard units) in relation to the native forces of the all-atom energy function called membrane_highres_Menv_smooth (*Alford et al., 2020*). The weight of the added constraints was 0.1 as not to overshadow the natural energetics of the protein system. The protein-membrane topology was predicted using TOPCONS (*Bernsel et al., 2009*), which was accounted for using the AddMembraneMover. To fold the protein with the modified energy function, energy minimization with a Monte-Carlo component was applied using the provided fastrelax algorithm with five repeats. The optimization was conducted in cartesian space (using a pro_close weight of 0.0 and cart_bonded weight of 0.5 as recommended for cartesian minimization), rather than in *Z*-matrix form such that the constraints could be applied correctly.

Additionally, to assess the quality of the predicted structures, the RMSD toward the structure of the closest available relative in the same conformational state was calculated. The closest available relative was determined by the BLOSUM62 distance (*Henikoff and Henikoff, 1992*) matrix, after filtering for the appropriate conformational state.

## MD simulations

All MD simulations were carried out in GROMACS2021 (*Abraham et al., 2015*). The simulation systems containing the predicted structures of GLUT5 in the outward-open, outward-occluded, inward-occluded, and inward-open states were prepared using the CHARMM-GUI membrane builder (*Jo et al., 2008*). The systems contained the protein, embedded in a POPC bilayer plunged in a 0.1 M KCl solution. The initial PBC box was 85 × 85 × 94 Å (*Reddy et al., 2012*), ensuring at least 12.5 Å of water molecules between the protein and the PBC box end at least 10 lipid molecules between each PBC copy of the protein. The force field used was CHARMM36m for protein and lipids (*Huang et al., 2017*), and TIP3P (*Jorgensen et al., 1983*) for water. The models were equilibrated using the default CHARMM-GUI scheme with one minimization step, and 6 100 ps restraint cycles with gradually released restraints in the NPT ensemble, followed by a production simulation of 10 ns. The simulations were carried out using a 2-fs time step. The target temperature and pressure were set to 303.15 K and 1 bar, respectively, and maintained by a Nose–Hoover thermostat (*Nosé, 1984*) (coupling separately protein, lipids, and solvent) and a Parrinello–Rahman barostat (*Parrinello and Rahman, 1981*) with semi-isotropic coupling (p = 5.0, compressibility 4.5 × 10$^{-5}$). Hydrogen bonds were constrained using the linear constraint solver (LINCS) (*Hess, 2008*), and long-range electrostatics were accounted for using the particle mesh Ewald method beyond the 12 Å electrostatic cutoff (*Darden, 1993*). A

neighbor-list cutoff was used for vdw interactions with $r_{vdw}$ = 12 Å and a switching function starting at 10 Å.

## CV determination

An SVM (**Cortes and Vapnik, 1995**) was trained on all predicted structures to separate adjacent states (outward occluded and outward open, outward occluded and inward occluded, and inward occluded and inward open) using a linear kernel. We avoided training on the occluded state given that only one sugar porter, *Pf*HT1, was resolved in that state, possibly representing a functional outlier. To also keep track of the transporter-specific features of each model, we performed PCA (**Pearson, 1901**) on the same training set (represented in the *y*-axis of **Figure 5—figure supplement 2**). Indeed, the same procedure on the AlphaFold (**Jumper et al., 2021**) input structures showed that the highest variance was in transporter specifity, not conformational states. Ideally, we sought to preserve the transporter-specific features while only switching conformational states.

The highest SVM coefficients (>0.193, as determined by the first gap in the histogram of coefficients) were divided into two separate components based on their sign. This yielded two CVs, with $CV_1$ describing contacts specific to state 1 and $CV_2$ describing contacts specific to state 2:

$$CV_1 = \frac{1}{\sum_{i=0}^{N} max(0,c_i)} \sum_{i=0}^{N} max\left(0, c_i\right) * x_i \tag{7}$$

$$CV_2 = \frac{1}{\sum_{i=0}^{N} min(0,c_i)} \sum_{i=0}^{N} -min\left(0, c_i\right) * x_i \tag{8}$$

where $x_i$ is the minimum distance between the two residues, and $c_i$ the value of the coefficient from the SVM of contact *i*. The normalization factor was included to facilitate interpretability and to avoid assertion failures within the MD code used. Given *CV* construction, a high value of $CV_j$ thus corresponds to a low amount contacts specific to state *j*.

## Accelerated weight histogram simulations

To reconstruct the complete conformational cycle of GLUT5, enhanced sampling simulations were run using the AWH method as implemented in GROMACS2021 (**Abraham et al., 2015**).

Three distinct AWH simulations were run to model each transition, using as CVs the relationships between distances described in **Equation 7 and 8**. These CVs were coupled to a reference coordinate $\xi(x)$ using a harmonic restraint function, using different force constants for each process (**Table 3**). The target distribution was chosen as uniform with a free-energy cutoff of 40–60 kJ/mol (**Table 3**) to avoid sampling of regions of high free energies. The free-energy estimate was updated each 100 steps, gathering data from every previous tenth step.

During the initial phase, the update bias size is first held constant (using a diffusion constant, as specified in **Table 3**, and an initial error of 10 kJ/mol), and then divided by three each time the CVs covers the entire target region. After the number of visits at each point grows larger than the histogram size at that point, the initial phase is exited, after which the update size is continuously and

**Table 3.** Input parameters used for the accelerated weighted histogram (AWH) simulations.
The force constant determines the resolution of the free-energy surface and the bias toward the reference coordinate. The cover diameter determines the radius in collective variable (CV) space that has to be covered before a walker shares the bias at that point. The free-energy cutoff determined at which level sampling is deemed uninteresting. The convergence time is the simulation time per walker that was needed to achieve convergence according to the stated criteria.

| | Diffusion constant (nm²/ps) | | Force constant (kJ/mol/ nm²) | | Cover diameter (nm) | | Free-energy cutoff (kJ/mol) | Convergence time (ns/walker) |
|---|---|---|---|---|---|---|---|---|
| | $CV_1$ | $CV_2$ | $CV_1$ | $CV_2$ | $CV_1$ | $CV_2$ | | |
| Outward opening | 0.0005 | 0.001 | 10,000 | 50,000 | 0.4 | 0.4 | 60 | 374 |
| Rocker switch | 0.001 | 0.001 | 50,000 | 50,000 | 0.4 | 0.4 | 60 | 653 |
| Inward opening | 0.0005 | 0.0005 | 30,000 | 30,000 | 0.4 | 0.4 | 40 | 235 |

exponentially decreased according to an exp-linear setting. All AWH simulations were run with four walkers in parallel with two starting in each conformational state at the extremes of each process.

Convergence of the AWH calculations was determined according to three criteria: the free-energy landscape was stable over time, the coordinate distribution along the minimum free-energy path reached a standard deviation of less than 0.1 from the mean coordinate distribution, and the changes in free-energy estimate were below 0.5 kJ/mol. The simulations were extended by 25 ns after these three criteria were met to ensure that these properties held true over time. The regions of high free energy (above the process-specific cutoff, see *Table 3*) of the free-energy landscape were excluded from the convergence analysis. To assess the convergence graphically, the 2-norm every 100 ps was calculated (*Figure 5—figure supplement 1*).

Since four independent walkers were used a coverage diameter of 4 Å was introduced, which forces each walker to explore an area of 4 Å around every point before sharing the bias with all other walkers. In practice, this results in walkers overlapping in phase space, which provides each point with a free-energy estimate based on multiple walkers. Effectively, this provides an inherent CV quality control for the method as different walkers should produce consistent free-energy estimates.

## Reweighting procedure

Seeing as all three processes (outward opening, rocker switch, and inward opening) were run independently, we needed a method to combine the results into one free-energy landscape, preferably along one dimension. To achieve this, we modified the canonical LDA loss function to include all adjacent state-wise terms:

$$L\left(\vec{w}, \vec{x}\right) = \frac{\sum_{i=0}^{n_c} \sum_{j=0}^{n_c} \left(i-j\right)^2 * \left(\mu_i - \mu_j\right)^2}{\sum_{i=0}^{n_c} \sum_{n=0}^{N_i} \left(\vec{w}^T x_n - \mu_i\right)^2} \tag{9}$$

where $n_c$ is the number of states, $N_i$ the number of points of state $i$, $w$ the coefficient vector, $x$ the dataset, and   the mean within each state. Superficially speaking, the loss function promotes a large mean difference between states, and a small standard deviation within each state. Importantly, this definition breaks the usually convex function of LDA, and is instead only locally convex in the regime where all state labels $i, j$ are correctly ordered. Due to this fact we ran the optimization 1000 times with random initializations and picked the coefficient vector that produced the correctly ordered and optimally divided the conformational states. Additionally, an *L2* regularization term was added with respect to the coefficient vector.

The function was then maximized with respect to the coefficient vector $w$ using the contact maps with the top 500 coevolving contacts of all predicted models as the dataset $x$. The implementation was made in Python3.8 using TensorFlow and is available in the Zenodo repository.

With an optimized CV at hand, we reweighted all samples obtained in the AWH simulations along $C\left(x_n\right) = \vec{w}^T x_n$ , by first calculating the weight of each frame in the simulation:

$$w\left(t\right) = e^{-U(\xi(t))/RT} / \sum_i^{N_{frames}} u\left(i, t, \xi\right) \tag{10}$$

where $U$ is the convolved free-energy estimate from AWH, $R$ the gas constant, $T$ 298 K and $u\left(i, t, \xi\right)$ is the binning procedure that is 1 if $\xi\left(i\right), \xi\left(t\right)$ are in the same bin, and 0 elsewhere. With these weights, the relative free energy of bin $i$ along a new coordinate $x_i$ can be calculated as:

$$E\left(x_i\right) = -RT \ln\left(\frac{\sum_{t=0}^{N_{frames}} u\left(i, t, x\right) * w\left(t\right)}{\sum_{t=0}^{N_{frames}} w\left(t\right)}\right) \tag{11}$$

The energy profile based on each individual simulation was then calculated, and aligned such that the free-energy shifts between sampled regions were accounted for. The aligning procedure relied on optimizing the free-energy shift to minimize the sum of distances between all points, weighted with the inverse of their standard error estimations. The resulting plot is seen in *Figure 5D*.

## Error analysis

To compare the quality of the different free-energy estimates, we sought a way to determine the point-wise uncertainty in the free-energy estimation. We estimated the point-wise error in the free-energy

surfaces by exploiting the fact that the goal of AWH simulations is to obtain a flat probability distribution. Our strategy was based on measuring the transition imbalance between neighboring bins in the 2D landscapes, which can then be used to estimate the standard error along the 1D free-energy landscape.

In adaptive bias simulations, the bias function $-U\left(\xi\left(r\right)\right)$ approximates the underlying Hamiltonian $H\left(\xi\left(r\right)\right)$, meaning that the effective Hamiltonian should produce a flat coordinate distribution:

$$P_{eff}\left(\xi\left(r\right)\right) = e^{U(\xi(r))-H(\xi(r))/RT} \tag{12}$$

In this work, we track the point-wise deviation from this distribution and calculate the error due to this with the following procedure. Given that all points have been sufficiently sampled several times (which is part of the convergence criteria in AWH), we can calculate the transition matrices between adjacent frames, and thus obtain the transition probabilities which can be translated into free-energy errors in a similar spirit as in Markov-state modeling (*Husic and Pande, 2018*). For an optimally converged system, all transitions should be roughly equally probable, and the higher the imbalance the higher the free-energy estimate error is in this region (*Figure 5—figure supplement 5*).

Additionally, it is important to quantify the sampling stemming from different walkers, and whether the free-energy estimate is consistent between different walkers. We did this by first calculating the unbiased coordinate distributions $P_i\left(x\right)$ for each walker $i$, and then defining the overlap metric for each bin $x$ as:

$$\mathrm{O}(x) = \sum_{i=0}^{N_{walkers}} \left\{ \begin{array}{l} 0, P_i\left(x\right) < 0.1 \\ 1, P_i\left(x\right) \geq 0.1 \end{array} \right\} \tag{13}$$

In a nutshell, the function describes how many walkers have a coordinate distribution of above 10% of the mean distribution at that position. Usually, this corresponds to a few thousand samples. The resulting distributions of $O(x)$ and $P(x)$ can be seen in *Figure 5—figure supplement 4*. The analysis shows that the majority of points are heavily visited by at least two to three walkers for all simulations, and only rarely, often in transition region do all walkers overlap. Importantly, it is in these transition regions that several walkers should overlap to produce a reliable estimate of the relative free energies between basins.

## Acknowledgements

This work was funded by the Knut and Alice Wallenberg Foundation (DD) and the Science for Life Laboratory (DD and LD), the Göran Gustafsson Foundation (DD and LD), and the Swedish Research Council (VR 2019-02433 to DD and LD). The MD simulations were performed on resources provided by the Swedish National Infrastructure for Computing (SNIC) on Beskow at the PDC Center for High Performance Computing (PDC-HPC).

## Additional information

### Competing interests
David Drew, Lucie Delemotte: Reviewing editor, *eLife*. The other authors declare that no competing interests exist.

### Funding

| Funder | Grant reference number | Author |
|---|---|---|
| Knut och Alice Wallenbergs Stiftelse | | David Drew |
| Science for Life Laboratory | | Lucie Delemotte David Drew |

| Funder | Grant reference number | Author |
|---|---|---|
| Gustafsson Foundation | | David Drew<br>Lucie Delemotte |
| Vetenskapsrådet | 2019-02433 | David Drew<br>Lucie Delemotte |

The funders had no role in study design, data collection, and interpretation, or the decision to submit the work for publication.

## Author contributions
Darko Mitrovic, Conceptualization, Data curation, Software, Formal analysis, Validation, Investigation, Visualization, Methodology, Writing - original draft, Writing - review and editing; Sarah E McComas, Claudia Alleva, Validation, Investigation, Visualization, Writing - review and editing; Marta Bonaccorsi, Validation, Investigation, Writing - review and editing; David Drew, Lucie Delemotte, Conceptualization, Resources, Supervision, Funding acquisition, Validation, Investigation, Writing - original draft, Project administration, Writing - review and editing

## Author ORCIDs
Darko Mitrovic ⓘ http://orcid.org/0000-0002-3219-1062
Claudia Alleva ⓘ http://orcid.org/0000-0001-8595-9250
David Drew ⓘ http://orcid.org/0000-0001-8866-6349
Lucie Delemotte ⓘ http://orcid.org/0000-0002-0828-3899

## Decision letter and Author response
Decision letter https://doi.org/10.7554/eLife.84805.sa1
Author response https://doi.org/10.7554/eLife.84805.sa2

# Additional files

## Supplementary files
• MDAR checklist

## Data availability
The data necessary to reproduce this work is openly available at https://doi.org/10.5281/zenodo.7899203.

The following dataset was generated:

| Author(s) | Year | Dataset title | Dataset URL | Database and Identifier |
|---|---|---|---|---|
| Mitrovic D | 2023 | Sugar transporter mechanism | https://doi.org/10.5281/zenodo.7899203 | Zenodo, 10.5281/zenodo.7899203 |

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
