## [Editor Report]

This important work proposes a novel approach, based on co-evolution analysis, machine-learning protocols, and molecular dynamics simulations, to predict structures and energetics of the main states of the alternating access cycle of a family of membrane transporters, the sugar porters. The approach is compelling, especially the application of co-evolution and Alphafold to generate accurate models in different conformational states of a given protein, and will be of interest to the membrane transport and computational modeling communities.

---

## [Decision Letter]

**Decision letter after peer review:**

Thank you for submitting your article "Reconstructing the transport cycle in the sugar porter superfamily using coevolution-powered machine learning" for consideration by *eLife*. Your article has been reviewed by 3 peer reviewers, including Randy B Stockbridge as the Reviewing Editor and Reviewer #1, and the evaluation has been overseen by Kenton Swartz as the Senior Editor. The following individual involved in review of your submission has agreed to reveal their identity: Krishna D Reddy (Reviewer #2).

Essential revisions:

The reviewers were very positive about the approach, and thought this was a potentially very valuable contribution to the field. They were especially enthusiastic about the application of co-evolution and Alphafold to generate highly accurate models in different conformational states, and the reviewers appreciated the extensive amount of modeling work. However, the reviewers also thought that the first point listed here was a considerable issue that could undermine important conclusions and would need to be fixed.

1. The most critical issue is that the procedure to calculate the free energy profile in Figure 5D appears to be fundamentally flawed. A properly calculated one-dimensional (1D) free energy profile must reflect the probability distribution of a single, well-defined collective variable and cannot generally be derived by stitching together 1D profiles from different variables. As the free energy landscape is an essential part of the manuscript, this will need to be corrected with extra analysis, providing the necessary information has been stored in the trajectories (see Kumar et al. J Comp Chem, 1992).

In addition, the reviewers identified several other "essential revisions" that they believe can be addressed in a revision:

2. The authors should soften their claims that this method is superior to homology modeling (without having done a quantitative analysis) and that this method is an order of magnitude faster than comparable methods (which has not been shown, and might be difficult to demonstrate).

3. The free energy landscapes in Figure 5A and Figure 5B do not seem consistent with each other, and the authors do not discuss whether errors in the calculations that may contribute to this. Upon discussion, the reviewers thought this might be in part due to confusing labeling of the minima. In order to clarify this point, it would be useful for the authors to quantitatively assess the differences by showing the relative probability (or free energy difference) between outward-occluded and occluded states from each of the two landscapes.

4. The reviewers also noted that the free energy landscapes in the companion paper also (*eLife*-84808) appeared significantly different. For example, the occluded state is a barrier in Figure 2E of the other work while looks essentially the most stable state in Figure 5A of this work and is again a barrier in Figure 5B. Also, the inward open state seems unstable in Figure 2E of the other work while there is a clear stable minimum in Figure 5C of this work. The authors should justify/discuss this.

5. Simulations started from predicted models tend to drift away from the native structure in the multi microseconds time domain, unless restraints are applied. the authors should show compelling evidence that the predicted models used in the simulation are of sufficiently high quality (Proteins 2012; 80:2071-2079). Therefore, the authors should show evidence that the predicted models used in the simulation are of sufficiently high quality, especially if the backbone RMSD deviates over 1 Å from the experimental structure.

6. As the free energy calculations are based on simulations started from different structures it would be useful to show free energy estimates from these individual simulations.

7. Since the quality of co-evolution analysis is largely dependent on the quality of the BLAST and sequence alignment, more detail regarding the methodology (trimming, manual editing, program parameters, sequence exclusion, etc.) is important to include. The actual sequence alignment and the list of proteins as a supplement should also be provided.

8. It is hard to fully conceptualize the extent of structural differences with 2D representations of aligned structures. A per-residue RMSD of various structure comparisons, mapped onto the experimentally solved structure, would help further illustrate the specific structural similarities and differences between the models and structures. This type of figure would be more helpful than the current Figures 4B and 4C.

*Reviewer #1 (Recommendations for the authors):*

Can you expand on the thought on page 15/line 22 that "even the rocker-switch bundle movement might utilize asymmetric rearrangements." I don't understand what the expectation for symmetry is.

*Reviewer #2 (Recommendations for the authors):*

– As the authors mention, most people use homology models to model specific conformations. Though I agree that the presented analysis is likely superior to such techniques, it is essential to demonstrate this quantitatively, both before and after MD simulations. Given the improvements post MD simulation, it would be interesting to see if the superior starting point (biased AlphaFold2 models) leads to a more improved final model, or if MD simulations are sufficient to approach the free energy minimum. This would further demonstrate the necessity of the described methodology and argue for its wider adoption.

– Since the quality of co-evolution analysis is largely dependent on the quality of the BLAST and sequence alignment, I would like some more detail regarding the methodology (trimming, manual editing, program parameters, sequence exclusion, etc.). The actual sequence alignment and the list of proteins as a supplement should also be provided. I was unable to find this in the provided OSF link; therefore, as it stands, I am not able to assess this data.

– It is hard to fully conceptualize the extent of structural differences with 2D representations of aligned structures. A per-residue RMSD of various structure comparisons, mapped onto the experimentally solved structure, would help further illustrate the specific structural similarities and differences between the models and structures. This type of figure would be more helpful than the current Figures 4B and 4C, as the improvements are hard to get a sense of as currently presented. Furthermore, this would answer the related question regarding if MD simulations improve RosettaMP models in specific ways, or is it a more global improvement.

– The proposed model of proton-coupling suggests conformation-specific pKa's of the aspartate residue, so that the transporter can bind protons in the outward-facing state, yet release protons in the inward-facing state. If this is the case, the co-evolution analysis should reveal residues adjacent (or perhaps even more allosteric) to the aspartate that could regulate the pKa in a conformation-dependent manner. This would be interesting to describe.

*Reviewer #3 (Recommendations for the authors):*

A critical issue of this work is that the procedure to calculate the free energy profile in Figure 5D is fundamentally flawed. A properly calculated one-dimensional (1D) free energy profile must reflect the probability distribution of a single, well-defined collective variable and cannot generally be derived by stitching together 1D profiles from different variables. Namely the orthogonal space of one variable, is generally not the same of that of another variable (and there are overlap regions between different variables). To do this correctly the authors should first define a single, mathematically well-defined variable describing the gradual structural variation from outward-open to inward-open conformations. A possibility for example is to use a path variable (J Chem Phys 2007 Feb 7;126(5):054103) derived from consecutive configurations from the three different paths. The authors could then use a reweighting approach to properly calculate the free energy along this path from the sampling of all simulations. To do this rigorously, the authors could use the weighted histogram analysis or the multistate Bennett acceptance ratio method, so that biases on different variables and overlap regions are properly accounted for.

The free energy landscapes in Figure 5A and Figure 5B do not seem consistent with each other. Namely while the occluded state is a main free energy minimum in the landscape of Figure 5A, it seems to be a barrier region in Figure 5B. To quantitatively assess this, it would be useful that the authors show the relative probability (or free energy difference) between outward-occluded and occluded states from each of the two landscapes. To do this the authors could define a unique descriptor to discriminate outward-occluded and occluded states (using the same descriptor for each landscape) and evaluate their probability. A simple way to do this, assuming the overall bias potential is only a function of CV1 and CV2, is to calculate the cumulative weight of each state. Where the weight of a simulation frame can be calculated as exp{-F(i)/kT}/N(i), where F(i) is the free energy as a function of CV1 and CV2 in a small bin of those variables assigned to that frame and N(i) is the number of simulation frames in that bin. This simple scheme could be also used to project the 2D landscape on a single variable, but weighted histogram analysis or the multistate Bennett acceptance are generally more rigorous methods in this regard.

As the free energy calculations are based on simulations (walkers) started from different (endpoints) structures it would be useful to show free energy estimates from these individual simulations. If this is not possible because they cover different portions of the space, the authors should show a metric of overlap, to make sure that individual simulations do not explore completely separated regions of the configurational space, thus leading to unreliable free energies.

Based also on the previous considerations, there is no evidence that the methodology proposed leads to one order of magnitude speed up compared to other methods and it would be generally difficult to demonstrate for these types of systems.

Another important point is that the simulations are based on the predicted models rather than on experimental structures. Previous systematic studies (see for example Proteins 2012; 80:2071-2079) underline how simulations started from predicted models tend to drift away from the native structure in the multi microseconds time domain, unless restraints are applied, which could help structural improvements (see also Protein Science 2015 25:19-29). Therefore, the authors should show compelling evidence that the predicted models used in the simulation are of sufficiently high quality, especially if the backbone RMSD deviates over 1 Å from the experimental structure. For example, by showing that convectional MD simulations started either from the models or from the X-ray structures are both stable and sample similar conformations (e.g. based on pairwise RMSD of both side chains and backbone).

The results of the modeling part of the work seem encouraging, nonetheless a suggestion for the authors is that, besides the RMSD distribution in Figure 4A they show analogous data for a descriptor that can better differentiate structural differences between states, as for example based on state-specific contacts. In particular, is not uncommon that the backbone RMSD between different states of a transporter is 3 Å or smaller.

---

## [Author Response]

Essential revisions:The reviewers were very positive about the approach, and thought this was a potentially very valuable contribution to the field. They were especially enthusiastic about the application of co-evolution and Alphafold to generate highly accurate models in different conformational states, and the reviewers appreciated the extensive amount of modeling work. However, the reviewers also thought that the first point listed here was a considerable issue that could undermine important conclusions and would need to be fixed.1. The most critical issue is that the procedure to calculate the free energy profile in Figure 5D appears to be fundamentally flawed. A properly calculated one-dimensional (1D) free energy profile must reflect the probability distribution of a single, well-defined collective variable and cannot generally be derived by stitching together 1D profiles from different variables. As the free energy landscape is an essential part of the manuscript, this will need to be corrected with extra analysis, providing the necessary information has been stored in the trajectories (see Kumar et al. J Comp Chem, 1992).

The original idea behind Figure 5D was to visualize the free energy landscape rather than provide a thermodynamic picture of the process, but we certainly agree that this did not come across properly in the submitted version, and that a thermodynamically robust reweighting scheme would lend quantitative insights to the work. In principle, a global collective variable along which all the states along the conformational cycle are separable would have to be defined and the probability distribution (free energy) projected along it. While maximum margin hyperplanes were defined for adjacent conformational states in the original work, we quickly found that they were incapable of separating all the states along the conformational cycle and thus to appropriately describe the global conformational change. Therefore, we modified the canonical linear discriminant analysis (LDA) equation for maximal separation of multiple labels and optimized it with respect to all predicted structures. Details of the methodology have been provided in the method section, and the procedure is described in the main text.

As mentioned by the reviewer, once the so-called LDA CV had been defined, we could use the thermodynamic weights from the AWH simulations to reweight the samples via the recommended reweighting scheme (using cumulative weights as exp{-F(i)/kT}/N(i)). The procedure is described in detail in Lindahl et. al, PRE 2018, and has been summarized in the methods section of the revised manuscript. Moreover, all three AWH simulations were projected onto the same space, and a WHAM-like procedure was used to determine the free energy shift between different profiles. In brief, the different free energy landscapes were aligned to maximize the statistical weight overlap between them. This methodology is described in the methods section (page 24-26)

Lastly, Figure 5 was redesigned to show the new projection, and the Results section has been updated accordingly (page 10-11).

In addition, the reviewers identified several other "essential revisions" that they believe can be addressed in a revision:2. The authors should soften their claims that this method is superior to homology modeling (without having done a quantitative analysis) and that this method is an order of magnitude faster than comparable methods (which has not been shown, and might be difficult to demonstrate).

We agree that no quantitative analysis has been made on using e.g. the method introduced herein versus MSM modelling or the swarms-of-strings method. However, we aimed to provide a rough estimate of the efficiency of the sampling that is achieved by the entire pipeline. Indeed, the same conformational change was tracked in the co-submitted paper by McComas et al. using the string-of-swarms method, leading us to believe that a comparison between these two methodologies is sufficient to suggest that the methodology provides a speed-up. Nevertheless we agree that a more comprehensive comparison would have to be made to provide the necessary quantifications. We have thus softened our claims to reflect this aspect. In addition, we have removed the claims about homology modelling completely given the difficulty in proving them, as correctly identified by the reviewers.

3. The free energy landscapes in Figure 5A and Figure 5B do not seem consistent with each other, and the authors do not discuss whether errors in the calculations that may contribute to this. Upon discussion, the reviewers thought this might be in part due to confusing labeling of the minima. In order to clarify this point, it would be useful for the authors to quantitatively assess the differences by showing the relative probability (or free energy difference) between outward-occluded and occluded states from each of the two landscapes.

We acknowledge that the original positioning of the labels led to confusion and have implemented several changes to better characterize the free energy landscapes and, in particular, the localization of their local minima. To address the differences between the projections in 2 and 1D, we added a section discussing the spaces where the free energy estimation from different simulations overlapped. This was made possible thanks to the reweighting onto the LDA CV (see point 1 in essential revisions above), which lends a common framework along which the relative free energy differences can be compared. We also wanted to quantify whether any disparities where because of lack of confidence in the free energy estimates, or due to systematic errors due to e.g. initialization of simulations from different starting points. In order to quantify this, we estimated the errors of the 2D AWH free energy landscapes by utilizing the fact that the probability distribution after converging AWH simulations should be flat, and quantified the energetic imbalance between bins to quantify the local error (see methods for details). With the previously mentioned reweighting procedure, we could estimate the standard error at each point along the LDA CV. These changes are mostly reflected in figure 5 but are thoroughly described in the methods sections (page 24-26).

To comment on the perceived disparity between figure 5A and 5B in particular, the rocker switch simulations between the outward occluded and the inward occluded states correctly assessed (within the standard error margin of ~1-2 kJ/mol) the difference between the outward occluded and occluded basins. To our surprise, these simulations even sampled the outward open states, despite the fact that the CVs were not designed to extend outside of the rocker switch conformational change, albeit with significantly higher standard errors. Notably, however, free energy estimates from this simulation in this region do not overlap with the estimate from the outward opening simulations. Since we did not target sampling of the outward opening process when designing the CV for the rocker-switch simulations, we surmise that there are strong reasons not to use this estimate of the outward open energy into consideration for analysis. This is discussed in the main text in relation to figure 5 (page 10-11).

4. The reviewers also noted that the free energy landscapes in the companion paper also (eLife-84808) appeared significantly different. For example, the occluded state is a barrier in Figure 2E of the other work while looks essentially the most stable state in Figure 5A of this work and is again a barrier in Figure 5B. Also, the inward open state seems unstable in Figure 2E of the other work while there is a clear stable minimum in Figure 5C of this work. The authors should justify/discuss this.

With the new projection into the LDA space, we could make more quantitative estimates of the free energies of the different states. The projection in the LDA space reveals that the concatenation of the free energy surfaces in the original submission led to a wrong positioning of the occluded state at a high free energy ~22 kJ/mol. As a result, there are no highly populated states in the inward-facing region, which is consistent with the companion paper.

Regarding the occluded state, we appreciate the reviewers’ comments and provide a more extensive explanation of the way we have characterized and described it. To do so, we have clarified both the definition and placement of the occluded state in multiple occasions in the revised manuscript. Firstly, we have clarified how we define the occluded state, following the nomenclature introduced by the Drew lab and adopted by the Yang lab, as an outward-facing state where the TM7b helix is broken (refer to revised Figure 1). Then, an analysis of the structural determinants of each metastable state (defined as conformational ensemble extracted from the free energy basins in Figure 5) is included in the supplementary information.

Secondly, by definition, the occluded state in GLUT5 must localize in a free energy basin (i.e. be a metastable state), and as close as possible in CV space to where the experimental structures (PfHT1 in the occluded state) fall. In the LDA space, the PfHT1 occluded state still localizes before the highest free energy barrier, while when the projection is carried out in the gate distance space of the companion paper, the PfHT1 occluded state falls directly on the barrier, highlighting the effect of choosing different low-dimensional CVs. Clearly, the occluded state in PfHT1 is a model of a transition state in GLUT5, while the corresponding thermodynamically stable occluded state is located prior to the rocker switch motion, consistent with the predominantly outward-facing nature of the occluded state. While this difference between transition state model and thermodynamically metastable state was mentioned in the original version of our manuscript, we have attempted to improve this discussion in this revised version of the manuscript.

In addition, an analysis has been provided to show that the labelling of the basin is indeed consistent with the definition in figure 1, which agrees with previous descriptions of the various conformational states, and in particular of the occluded state as described in structural work on PfHT1. (supporting information Figure 5 - figure supplement 6 and page 10 in main text)

5. Simulations started from predicted models tend to drift away from the native structure in the multi microseconds time domain, unless restraints are applied. the authors should show compelling evidence that the predicted models used in the simulation are of sufficiently high quality (Proteins 2012; 80:2071-2079). Therefore, the authors should show evidence that the predicted models used in the simulation are of sufficiently high quality, especially if the backbone RMSD deviates over 1 Å from the experimental structure.

To assess the stability of the models over time, we calculated the heavy-atom RMSD of the extended simulations starting in the GLUT5 predicted states, namely the outward open, outward occluded, inward occluded and inward open models. While most simulations are within the expected range of stability and exhibit little change over the majority of the trajectory, the outward open model seems to fluctuate the most. Since the nature of these fluctuations is relatively discontinuous, we explored whether alternative conformational states are explored. Indeed, while the RMSD is as high as 5-6Å to the starting outward open model, it is significantly lower, between 1.5-2.5Å to the outward occluded model, suggesting that over these timescales, it is possible to observe a conformational change towards the outward occluded state. In addition, projecting the time-evolution of these simulations in LDA CV space paints a similar picture, consistent with the slope of the 1d free energy landscape in this region (Figure 5 - figure supplement 7). Figures have been added in the supporting information to support these statements, and the text describing these features has been expanded (page 11)

6. As the free energy calculations are based on simulations started from different structures it would be useful to show free energy estimates from these individual simulations.

We disagree with the above and find that it is not very informative to show free energy estimates from the individual simulations. In multiple walker AWH, the global free energy estimate is not easily decoupled from each individual walker, but is based on the compounded distribution and response to the adaptive bias. Thus, if we restrict the free energy estimate to a single walker we will simply get the same free energy surface in the space where the given walker has sampled.

In contrast, to attempt to follow the reviewer’s recommendation to directly report on the sampling from each walker, we calculated the overlap between regions sampled by each walker. To do so, we quantified the overlap by calculating bin-wise the number of walkers whose coordinate distribution exceeds 10%. This is shown in the supporting information (Figure 5 - figure supplement 4,5) and gives a clear picture, along with the calculated errors, of the regions of the free energy surface that can be trusted and the ones which have lower confidence.

7. Since the quality of co-evolution analysis is largely dependent on the quality of the BLAST and sequence alignment, more detail regarding the methodology (trimming, manual editing, program parameters, sequence exclusion, etc.) is important to include. The actual sequence alignment and the list of proteins as a supplement should also be provided.

Following the reviewer’s comments, we have expanded the methods section and provided the necessary details (page 18). We have also uploaded the MSA used for coevolution to our open access Zenodo directory. We have also clarified which parts of the datasets were actually used for the coevolution analysis, and which were omitted.

8. It is hard to fully conceptualize the extent of structural differences with 2D representations of aligned structures. A per-residue RMSD of various structure comparisons, mapped onto the experimentally solved structure, would help further illustrate the specific structural similarities and differences between the models and structures. This type of figure would be more helpful than the current Figures 4B and 4C.

While the 3D representations can help with contextualizing which regions were subject to the largest disparities, we enthusiastically agree that an RMSD plot would be beneficial to show quantitative differences. Following this suggestion, we added RMSD plots of the shown structures and labelled the relevant regions for comparison, as done in the 3D structures. (Figure 4B-C)

Reviewer #1 (Recommendations for the authors):Can you expand on the thought on page 15/line 22 that "even the rocker-switch bundle movement might utilize asymmetric rearrangements." I don't understand what the expectation for symmetry is.

We expanded on the rocker switch mechanism generally at play in the well-studied members of the MFS-transporter family. For example, in SWEETs, the substrate translocation is significantly more symmetric than in GLUTs. (page 17)

Reviewer #2 (Recommendations for the authors):– As the authors mention, most people use homology models to model specific conformations. Though I agree that the presented analysis is likely superior to such techniques, it is essential to demonstrate this quantitatively, both before and after MD simulations. Given the improvements post MD simulation, it would be interesting to see if the superior starting point (biased AlphaFold2 models) leads to a more improved final model, or if MD simulations are sufficient to approach the free energy minimum. This would further demonstrate the necessity of the described methodology and argue for its wider adoption.

As the reviewer mentions, it is difficult to provide a quantification of any potential improvement compared to other structure prediction methods such as homology modelling without systematically predicting all presented structures using homology modelling and comparing them to a ground truth. The necessary quantifications would thus require a wider study with different templates and an unambiguous estimate of transporter-specific features. While outside of the scope of this article, there are as of yet no set of transporter-specific features to validate against due to the lack of experimental structures of the solved transporters in all states.

– The proposed model of proton-coupling suggests conformation-specific pKa's of the aspartate residue, so that the transporter can bind protons in the outward-facing state, yet release protons in the inward-facing state. If this is the case, the co-evolution analysis should reveal residues adjacent (or perhaps even more allosteric) to the aspartate that could regulate the pKa in a conformation-dependent manner. This would be interesting to describe.

The coevolution analysis reveals all evolutionarily conserved contacts (more specifically, coupled residues) in the protein, whereas uncovering the state-specific function of each contact required the more extended analysis presented in the paper. To determine if and how there is an evolutionary record of a proton chain in this family would require a separate analysis that we deem is beyond the scope of this work, albeit very interesting. In fact, it is not even known whether a proton chain is necessary, or if deprotonation can occur at the same site when the protein is in the inward-facing state. A more focused study on e.g. XylE, a known proton-coupled transporter and relatively well-characterized protein would be more appropriate in this regard. We thus added this proposition in the outlook section of the paper. (page 16)

Reviewer #3 (Recommendations for the authors):The results of the modeling part of the work seem encouraging, nonetheless a suggestion for the authors is that, besides the RMSD distribution in Figure 4A they show analogous data for a descriptor that can better differentiate structural differences between states, as for example based on state-specific contacts. In particular, is not uncommon that the backbone RMSD between different states of a transporter is 3 Å or smaller.

We agree with the reviewer’s assessment that other structural descriptors can help characterize the inferred states. While the description of state-specific contacts for all states were introduced as input in this work, and thus cannot be used for validation, we instead characterize the distribution of commonly agreed-upon observables. We now present these early in the renewed version of the main text, having added a graphic explaining these observables in figure 1. We then analyze the conformational ensembles making up free energy minima with respect to these observables, contrasting them with reference values for each structurally-determined state. The definitions are described in detail in the methods section and the text and figures have been modified. (page 3-4, 10-11, figure 1 and Figure 5 - Figures Supplement 6).